# Transformers are Deep Infinite-Dimensional Non-Mercer Binary Kernel Machines

## Abstract

Despite their ubiquity in core AI fields like natural language processing, the mechanics of deep attention-based neural networks like the "Transformer" model are not fully understood. In this article, we present a new perspective towards understanding how Transformers work. In particular, we show that the "dot-product attention" that is the core of the Transformer's operation can be characterized as a kernel learning method on a pair of Banach spaces. In particular, the Transformer's kernel is characterized as having an infinite feature dimension. Along the way we generalize the standard kernel learning problem to what we term a "binary" kernel learning problem, where data come from two input domains and a response is defined for every cross-domain pair. We prove a new representer theorem for these binary kernel machines with non-Mercer (indefinite, asymmetric) kernels (implying that the functions learned are elements of reproducing kernel Banach spaces rather than Hilbert spaces), and also prove a new universal approximation theorem showing that the Transformer calculation can learn any binary non-Mercer reproducing kernel Banach space pair. We experiment with new kernels in Transformers, and obtain results that suggest the infinite dimensionality of the standard Transformer kernel is partially responsible for its performance. This paper's results provide a new theoretical understanding of a very important but poorly understood model in modern machine learning.

## 1 Introduction

Since its proposal by Bahdanau et al. (2015), so-called neural attention has become the backbone of many state-of-the-art deep learning models. This is true in particular in natural language processing (NLP), where the Transformer model of Vaswani et al. (2017) has become ubiquitous. This ubiquity is such that much of the last few years' NLP breakthroughs have been due to developing new training regimes for Transformers (Radford et al., 2018; Devlin et al., 2019; Yang et al., 2019; Liu et al., 2019; Wang et al., 2019a; Joshi et al., 2020; Lan et al., 2020; Brown et al., 2020, etc.).

Like most modern deep neural networks, theoretical understanding of the Transformer has lagged behind the rate of Transformer-based performance improvements on AI tasks like NLP. Recently, several authors have noted Transformer operations' relationship to other, better-understood topics in deep learning theory, like the similarities between attention and convolution (Ramachandran et al., 2019; Cordonnier et al., 2020) and the design of the residual blocks in multi-layer Transformers (e.g., Lu et al. (2019); see also the reordering of the main learned (fully-connected or attentional) operation, elementwise nonlinearity, and normalization in the original Transformer authors' official reference codebase (Vaswani et al., 2018) and in some more recent studies of deeper Transformers (Wang et al., 2019b) to the "pre-norm" ordering of normalize, learned operation, nonlinearity, add residual ordering of modern ("v2") Resnets (He et al., 2016)).

In this paper, we propose a new lens to understand the central component of the Transformer, its "dot-product attention" operation. In particular, we show dot-product attention can be characterized as a particular class of kernel method (Schölkopf & Smola, 2002). More specifically, it is a so-called *indefinite* and *asymmetric* kernel method, which refer to two separate generalizations of the classic class of kernels that does not require the classic assumptions of symmetry and positive (semi-) definiteness (Ong et al., 2004; Balcan et al., 2008; Zhang et al., 2009; Wu et al., 2010; Loosli et al.,

2016; Oglic & Gärtner, 2018; 2019, etc.). We in fact show in Theorem 2 below that dot-product attention can learn any asymmetric indefinite kernel.

This insight has several interesting implications. Most immediately, it provides some theoretical justification for one of the more mysterious components of the Transformer model. It also potentially opens the door for the application of decades of classic kernel method theory towards understanding one of today's most important neural network models, perhaps similarly to how tools from digital signal processing are widely used to study convolutional neural networks. We make a first step on this last point in this paper, proposing a generalization of prior kernel methods we call "binary" kernel machines, that learns how to predict distinct values for pairs of elements across two input sets, similar to an attention model.

The remainder of this paper is organized as follows. Section 2 reviews the mathematical background of both Transformers and classic kernel methods. Section 3 presents the definition of kernel machines on reproducing kernel Banach spaces (RKBS's) that we use to characterize Transformers. In particular we note that the Transformer can be described as having an infinite-dimensional feature space. Section 4 begins our theoretical results, explicitly describing the Transformer in terms of reproducing kernels, including explicit formulations of the Transformer's kernel feature maps and its relation to prior kernels. Section 5 discusses Transformers as kernel learners, including a new representer theorem and a characterization of stochastic-gradient-descent-trained attention networks as approximate kernel learners. In Section 6, we present empirical evidence that the infinite-dimensional character of the Transformer kernel may be somewhat responsible for the model's effectiveness. Section 7 concludes and summarizes our work.

## 2 BACKGROUND AND RELATED WORK

### 2.1 TRANSFORMER NEURAL NETWORK MODELS

The Transformer model (Vaswani et al., 2017) has become ubiquitous in many core AI applications like natural language processing. Here, we review its core components. Say we have two ordered sets of vectors, a set of "source" elements $\{s_1, s_2, \ldots, s_S\}$, $s_j \in \mathbb{R}^{d_s}$ and a set of "target" elements $\{t_1, t_2, \ldots, t_T\}$, $t_i \in \mathbb{R}^{d_t}$. In its most general form, the neural-network "attention" operation that forms the backbone of the Transformer model is to compute, for each $t_i$, a $t_i$-specific embedding of the source sequence $\{s_j\}_{j=1}^S$.[1]

The particular function used in the Transformer is the so-called "scaled dot-product" attention, which takes the form

$$a_{ij} = \frac{(W^Q t_i)^\mathsf{T} (W^K s_j)}{\sqrt{d}} \qquad \alpha_{ij} = \frac{\exp(a_{ij})}{\sum_{j=1}^S \exp(a_{ij})} \qquad t_i' = \sum_{j=1}^S \alpha_{ij} W^V s_j \qquad (1)$$

where $W^V, W^K \in \mathbb{R}^{d_s \times d}$, and $W^Q \in \mathbb{R}^{d_t \times d}$ are learnable weight matrices, usually called the "value," "key," and "query" weight matrices, respectively. Usually multiple so-called "attention heads" with independent parameter matrices implement several parallel computations of (1), with the Cartesian product (vector concatenation) of several $d$-dimensional head outputs forming the final output $t_i'$. Usually the unnormalized $a_{ij}$'s are called attention scores or attention logits, and the normalized $\alpha_{ij}$'s are called attention weights.

In this paper, we restrict our focus to the dot-product formulation of attention shown in (1). Several other alternative forms of attention that perform roughly the same function (i.e., mapping from $\mathbb{R}^{d_s} \times \mathbb{R}^{d_t}$ to $\mathbb{R}$) have been proposed (Bahdanau et al., 2015; Luong et al., 2015; Veličković et al., 2018; Battaglia et al., 2018, etc.) but the dot-product formulation of the Transformer is by far the most popular.

### 2.2 KERNEL METHODS AND GENERALIZATIONS OF KERNELS

Kernel methods (Schölkopf & Smola, 2002; Steinwart & Christmann, 2008, etc.) are a classic and powerful class of machine learning methods. The key component of kernel methods are the namesake

---

[1]Often, the source and target sets are taken to be the same, $s_i = t_i \ \forall i$. This instance of attention is called self attention.

*kernel functions*, which allow the efficient mapping of input data from a low-dimensional data domain, where linear solutions to problems like classification or regression may not be possible, to a high- or infinite-dimensional embedding domain, where linear solutions can be found.

Given two nonempty sets $\mathcal{X}$ and $\mathcal{Y}$, a kernel function $\kappa$ is a continuous function $\kappa : \mathcal{X} \times \mathcal{Y} \to \mathbb{R}$. In the next few sections, we will review the classic symmetric and positive (semi-) definite, or Mercer, kernels, then discuss more general forms.

### 2.2.1 SYMMETRIC AND POSITIVE SEMIDEFINITE (MERCER) KERNELS

If $\mathcal{X} = \mathcal{Y}$, and for all $x_i, x_j \in \mathcal{X} = \mathcal{Y}$, a particular kernel $\kappa$ has the properties

$$\text{symmetry: } \kappa(x_i, x_j) = \kappa(x_j, x_i) \tag{2a}$$

$$\text{positive (semi-) definiteness: } \boldsymbol{c}^\mathsf{T} \boldsymbol{K} \boldsymbol{c} \geq 0 \quad \forall \, \boldsymbol{c} \in \mathbb{R}^n; \quad i, j = 1, \ldots, n; \quad n \in \mathbb{N} \tag{2b}$$

where $\boldsymbol{K}$ in (2b) is the Gram matrix, defined as $K_{ij} = \kappa(x_i, x_j)$, then $\kappa$ is said to be a Mercer kernel. For Mercer kernels, it is well-known that, among other facts, (i) we can define a Hilbert space of functions on $\mathcal{X}$, denoted $\mathcal{H}_\kappa$ (called the reproducing kernel Hilbert space, or RKHS, associated with the reproducing kernel $\kappa$), (ii) $\mathcal{H}_\kappa$ has for each $x$ a (continuous) unique element $\delta_x$ called a point evaluation functional, with the property $f(x) = \delta_x(f) \, \forall f \in \mathcal{H}_\kappa$, (iii) $\kappa$ has the so-called *reproducing property*, $\langle f, \kappa(x, \cdot) \rangle_{\mathcal{H}_\kappa} = f(x) \, \forall f \in \mathcal{H}_\kappa$, where $\langle \cdot, \cdot \rangle_{\mathcal{H}_\kappa}$ is the inner product on $\mathcal{H}_\kappa$, and (iv) we can define a "feature map" $\Phi : \mathcal{X} \to \mathcal{F}_\mathcal{H}$, where $\mathcal{F}_\mathcal{H}$ is another Hilbert space sometimes called the feature space, and $\kappa(x, y) = \langle \Phi(x), \Phi(y) \rangle_{\mathcal{F}_\mathcal{H}}$ (where $\langle \cdot, \cdot \rangle_{\mathcal{F}_\mathcal{H}}$ is the inner product associated with $\mathcal{F}_\mathcal{H}$). This last point gives rise to the kernel trick for RKHS's.

From a machine learning and optimization perspective, kernels that are symmetric and positive (semi-) definite (PSD) are desirable because those properties guarantee that empirical-risk-minimization kernel learning problems like support vector machines (SVMs), Gaussian processes, etc. are convex. Convexity gives appealing guarantees for the tractability of a learning problem and optimality of solutions.

### 2.2.2 LEARNING WITH NON-MERCER KERNELS

Learning methods with non-Mercer kernels, or kernels that relax the assumptions (2), have been studied for some time. One line of work (Lin & Lin, 2003; Ong et al., 2004; Chen & Ye, 2008; Luss & D'aspremont, 2008; Alabdulmohsin et al., 2015; Loosli et al., 2016; Oglic & Gärtner, 2018; 2019, etc.) has focused on learning with symmetric but indefinite kernels, i.e., kernels that do not satisfy (2b). Indefinite kernels have been identified as reproducing kernels for so-called reproducing kernel Kreĭn spaces (RKKS's) since Schwartz (1964) and Alpay (1991).

Replacing a Mercer kernel in a learning problem like an SVM with an indefinite kernel makes the optimization problem nonconvex in general (as the kernel Gram matrix $\boldsymbol{K}$ is no longer always PSD). Some early work in learning with indefinite kernels tried to ameliorate this problem by modifying the spectrum of the Gram matrix such that it again becomes PSD (e.g., Graepel et al., 1998; Roth et al., 2003; Wu et al., 2005). More recently, Loosli et al. (2016); Oglic & Gärtner (2018), among others, have proposed optimization procedures to learn in the RKKS directly. They report better performance on some learning problems when using indefinite kernels than either popular Mercer kernels or spectrally-modified indefinite kernels, suggesting that sacrificing convexity can empirically give a performance boost. This conclusion is of course reminiscent of the concurrent experience of deep neural networks, which are hard to optimize due to their high degree of non-convexity, yet give superior performance to many other methods.

Another line of work has explored the application of kernel methods to learning in more general Banach spaces, i.e., reproducing kernel Banach spaces (RKBS's) (Zhang et al., 2009). Various constructions to serve as the reproducing kernel for a Banach space (replacing the inner product of an RKHS) have been proposed, including semi-inner products (Zhang et al., 2009), positive-definite bilinear forms via a Fourier transform construction (Fasshauer et al., 2015), and others (Song et al., 2013; Georgiev et al., 2014, etc.). In this work, we consider RKBS's whose kernels may be neither symmetric nor PSD. A definition of these spaces is presented next.

## 3 GENERAL REPRODUCING KERNEL BANACH SPACES

Recently, Georgiev et al. (2014), Lin et al. (2019), and Xu & Ye (2019) proposed similar definitions and constructions of RKBS's and their reproducing kernels meant to encompass prior definitions. In this paper, we adopt a fusion of the definitions and attempt to keep the notation as simple as possible to be sufficient for our purposes.

**Definition 1** (**Reproducing kernel Banach space** (Xu & Ye, 2019, Definition 2.1; Lin et al., 2019, Definitions 1.1 & 1.2; Georgiev et al., 2014)). Let $\mathcal{X}$ and $\mathcal{Y}$ be nonempty sets, $\kappa$ a measurable function called a kernel, $\kappa : \mathcal{X} \times \mathcal{Y} \to \mathbb{R}$, and $\mathcal{B}_{\mathcal{X}}$ and $\mathcal{B}_{\mathcal{Y}}$ Banach spaces of real measurable functions on $\mathcal{X}$ and $\mathcal{Y}$, respectively. Let $\langle \cdot, \cdot \rangle_{\mathcal{B}_{\mathcal{X}} \times \mathcal{B}_{\mathcal{Y}}} : \mathcal{B}_{\mathcal{X}} \times \mathcal{B}_{\mathcal{Y}} \to \mathbb{R}$ be a nondegenerate bilinear mapping such that

$$\kappa(x, \cdot) \in \mathcal{B}_{\mathcal{Y}} \qquad \text{for all } x \in \mathcal{X}; \tag{3a}$$

$$\langle f, \kappa(x, \cdot) \rangle_{\mathcal{B}_{\mathcal{X}} \times \mathcal{B}_{\mathcal{Y}}} = f(x) \quad \text{for all } x \in \mathcal{X}, f \in \mathcal{B}_{\mathcal{X}}; \tag{3b}$$

$$\kappa(\cdot, y) \in \mathcal{B}_{\mathcal{X}} \qquad \text{for all } y \in \mathcal{Y}; \text{ and} \tag{3c}$$

$$\langle \kappa(\cdot, y), g \rangle_{\mathcal{B}_{\mathcal{X}} \times \mathcal{B}_{\mathcal{Y}}} = g(y) \quad \text{for all } y \in \mathcal{Y}, g \in \mathcal{B}_{\mathcal{Y}}. \tag{3d}$$

Then, $\mathcal{B}_{\mathcal{X}}$ and $\mathcal{B}_{\mathcal{Y}}$ are a pair of reproducing kernel Banach spaces (RKBS's) on $\mathcal{X}$ and $\mathcal{Y}$, respectively, and $\kappa$ is their reproducing kernel.

Line (3a) (resp. (3c)) says that, if we take $\kappa$, a function of two variables $x \in \mathcal{X}$ and $y \in \mathcal{Y}$, and fix $x$ (resp. $y$), then we get a function of one variable. This function of one variable must be an element of $\mathcal{B}_{\mathcal{Y}}$ (resp. $\mathcal{B}_{\mathcal{X}}$). Lines (3b) and (3d) are the *reproducing properties* of $\kappa$.

For our purposes, it will be useful to extend this definition to include a "feature map" characterization similar to the one used in some explanations of RKHS's (Schölkopf & Smola, 2002, Chapter 2).

**Definition 2** (**Feature maps for RKBS's** (Lin et al., 2019, Theorem 2.1; Georgiev et al., 2014)). For a pair of RKBS's as defined in Definition 1, suppose that there exist mappings $\Phi_{\mathcal{X}} : \mathcal{X} \to \mathcal{F}_{\mathcal{X}}, \Phi_{\mathcal{Y}} : \mathcal{Y} \to \mathcal{F}_{\mathcal{Y}}$, where $\mathcal{F}_{\mathcal{X}}$ and $\mathcal{F}_{\mathcal{Y}}$ are Banach spaces we will call the feature spaces, and a nondegenerate bilinear mapping $\langle \cdot, \cdot \rangle_{\mathcal{F}_{\mathcal{X}} \times \mathcal{F}_{\mathcal{Y}}} : \mathcal{F}_{\mathcal{X}} \times \mathcal{F}_{\mathcal{Y}} \to \mathbb{R}$ such that

$$\kappa(x, y) = \langle \Phi_{\mathcal{X}}(x), \Phi_{\mathcal{Y}}(y) \rangle_{\mathcal{F}_{\mathcal{X}} \times \mathcal{F}_{\mathcal{Y}}} \quad \text{for all } x \in \mathcal{X}, y \in \mathcal{Y}. \tag{4}$$

In this case, the spaces $\mathcal{B}_{\mathcal{X}}$ and $\mathcal{B}_{\mathcal{Y}}$ can be defined as (Xu & Ye, 2019; Lin et al., 2019)

$$\mathcal{B}_{\mathcal{X}} = \left\{ f_v : \mathcal{X} \to \mathbb{R} : f_v(x) \triangleq \langle \Phi_{\mathcal{X}}(x), v \rangle_{\mathcal{F}_{\mathcal{X}} \times \mathcal{F}_{\mathcal{Y}}} \; ; v \in \mathcal{F}_{\mathcal{Y}}, x \in \mathcal{X} \right\} \tag{5a}$$

$$\mathcal{B}_{\mathcal{Y}} = \left\{ g_u : \mathcal{Y} \to \mathbb{R} : g_u(y) \triangleq \langle u, \Phi_{\mathcal{Y}}(y) \rangle_{\mathcal{F}_{\mathcal{X}} \times \mathcal{F}_{\mathcal{Y}}} \; ; u \in \mathcal{F}_{\mathcal{X}}, y \in \mathcal{Y} \right\}. \tag{5b}$$

*Remark* 1. We briefly discuss how to understand the spaces given by (5). Consider (5a) for example. It is a space of real-valued functions of one variable $x$, where the function is also parameterized by a $v$. Picking a $v \in \mathcal{F}_{\mathcal{Y}}$ in (5a) defines a manifold of functions in $\mathcal{B}_{\mathcal{X}}$. This manifold of functions with fixed $v$ varies with the function $\Phi_{\mathcal{X}}$. Evaluating a function $f_v$ in this manifold at a point $x$ is defined by taking the bilinear product of $\Phi_{\mathcal{X}}(x)$ and the chosen $v$. This also means that we can combine (4) and (5) to say

$$\kappa(x, y) = \langle \Phi_{\mathcal{X}}(x), \Phi_{\mathcal{Y}}(y) \rangle_{\mathcal{F}_{\mathcal{X}} \times \mathcal{F}_{\mathcal{Y}}} = \langle f_{\Phi_{\mathcal{X}}(x)}, g_{\Phi_{\mathcal{Y}}(y)} \rangle_{\mathcal{B}_{\mathcal{X}} \times \mathcal{B}_{\mathcal{Y}}} \quad \text{for all } x \in \mathcal{X}, y \in \mathcal{Y}. \tag{6}$$

*Remark* 2. If $\Phi_{\mathcal{X}}(x)$ and $\Phi_{\mathcal{Y}}(y)$ can be represented as countable sets of real-valued measurable functions, $\{\phi_{\mathcal{X}}(x)_\ell\}_{\ell \in \mathbb{N}}$ and $\{\phi_{\mathcal{Y}}(y)_\ell\}_{\ell \in \mathbb{N}}$ for $(\phi_{\mathcal{X}})_\ell : \mathcal{X} \to \mathbb{R}$ and $(\phi_{\mathcal{Y}})_\ell : \mathcal{Y} \to \mathbb{R}$ (i.e., $\mathcal{F}_{\mathcal{X}}, \mathcal{F}_{\mathcal{Y}} \subset \prod_{\ell \in \mathbb{N}} \mathbb{R}$); and $\langle \boldsymbol{u}, \boldsymbol{v} \rangle_{\mathcal{F}_{\mathcal{X}} \times \mathcal{F}_{\mathcal{Y}}} = \sum_{\ell \in \mathbb{N}} u_\ell v_\ell$ for $\boldsymbol{u} \in \mathcal{F}_{\mathcal{X}}, \boldsymbol{v} \in \mathcal{F}_{\mathcal{Y}}$; then the "feature map" construction, whose notation we borrow from Lin et al. (2019), corresponds to the "generalized Mercer kernels" of Xu & Ye (2019).

## 4 DOT-PRODUCT ATTENTION AS AN RKBS KERNEL

We now formally state the formulation for dot-product attention as an RKBS learner. Much like with RKHS's, for a given kernel and its associated RKBS pair, the feature maps (and also the bilinear mapping) are not unique. In the following, we present a feature map based on classic characterizations of other kernels such as RBF kernels (e.g., Steinwart et al. (2006)).

**Proposition 1.** *The (scaled) dot-product attention calculation of* (1) *is a reproducing kernel for an RKBS in the sense of Definitions 1 and 2, with the input sets $\mathcal{X}$ and $\mathcal{Y}$ being the vector spaces from which the target elements $\{\boldsymbol{t}_i\}_{i=1}^{T}, \boldsymbol{t}_i \in \mathbb{R}^{d_t}$ and source elements $\{\boldsymbol{s}_j\}_{j=1}^{S}, \boldsymbol{s}_j \in \mathbb{R}^{d_s}$ are drawn, respectively; the feature maps*

$$\Phi_{\mathcal{X}}(\boldsymbol{t}) = \sum_{n=0}^{\infty} \sum_{p_1+p_2+\cdots+p_d=n} d^{-1/4} \left( \frac{n!}{p_1! p_2! \cdots p_d!} \right)^{1/2} \prod_{\ell=1}^{d} (q_\ell)^{p_\ell} \tag{7a}$$

$$\Phi_{\mathcal{Y}}(\boldsymbol{s}) = \sum_{n=0}^{\infty} \sum_{p_1+p_2+\cdots+p_d=n} d^{-1/4} \left( \frac{n!}{p_1! p_2! \cdots p_d!} \right)^{1/2} \prod_{\ell=1}^{d} (k_\ell)^{p_\ell} \tag{7b}$$

*where $q_\ell$ is the $\ell$th element of $\boldsymbol{q} = \boldsymbol{W}^Q \boldsymbol{t}$, $k_\ell$ is the $\ell$th element of $\boldsymbol{k} = \boldsymbol{W}^K \boldsymbol{s}$, $\boldsymbol{W}^Q \in \mathbb{R}^{d \times d_t}$ $\boldsymbol{W}^K \in \mathbb{R}^{d \times d_s}$, with $d \leq d_s, d_t$ and $\mathrm{rank}(\boldsymbol{W}^Q) = \mathrm{rank}(\boldsymbol{W}^K) = d$; the bilinear mapping $\langle \Phi_{\mathcal{X}}(\boldsymbol{t}), \Phi_{\mathcal{Y}}(\boldsymbol{s}) \rangle_{\mathcal{F}_{\mathcal{X}} \times \mathcal{F}_{\mathcal{Y}}} = \Phi_{\mathcal{X}}(\boldsymbol{t}) \cdot \Phi_{\mathcal{Y}}(\boldsymbol{s})$; and the Banach spaces*

$$\mathcal{B}_{\mathcal{X}} = \left\{ f_{\boldsymbol{k}}(\boldsymbol{t}) = \exp\left( (\boldsymbol{W}^Q \boldsymbol{t})^\mathsf{T} \boldsymbol{k} / \sqrt{d} \right) \ ; \ \boldsymbol{k} \in \mathcal{F}_{\mathcal{Y}}, \boldsymbol{t} \in \mathcal{X} \right\} \tag{8a}$$

$$\mathcal{B}_{\mathcal{Y}} = \left\{ g_{\boldsymbol{q}}(\boldsymbol{s}) = \exp\left( \boldsymbol{q}^\mathsf{T} (\boldsymbol{W}^K \boldsymbol{s}) / \sqrt{d} \right) \ ; \ \boldsymbol{q} \in \mathcal{F}_{\mathcal{X}}, \boldsymbol{s} \in \mathcal{Y} \right\} \tag{8b}$$

*with the "exponentiated query-key kernel,"*

$$\kappa(\boldsymbol{t}, \boldsymbol{s}) = \langle \Phi_{\mathcal{X}}(\boldsymbol{t}), \Phi_{\mathcal{Y}}(\boldsymbol{s}) \rangle_{\mathcal{F}_{\mathcal{X}} \times \mathcal{F}_{\mathcal{Y}}} = \langle f_{\Phi_{\mathcal{Y}}(\boldsymbol{s})}, g_{\Phi_{\mathcal{X}}(\boldsymbol{t})} \rangle_{\mathcal{B}_{\mathcal{X}} \times \mathcal{B}_{\mathcal{Y}}} = \exp\left( \frac{(\boldsymbol{W}^Q \boldsymbol{t})^\mathsf{T} (\boldsymbol{W}^K \boldsymbol{s})}{\sqrt{d}} \right) \tag{9}$$

*the associated reproducing kernel.*

The proof of Proposition 1 is straightforward and involves verifying (9) by multiplying the two infinite series in (7), then using the multinomial theorem and the Taylor expansion of the exponential.

In the above and when referring to Transformer-type models in particular rather than RKBS's in general, we use $\boldsymbol{t}, \boldsymbol{s}, \boldsymbol{q}$, and $\boldsymbol{k}$ for $x, y, u$, and $v$, respectively, to draw the connection between the elements of the RKBS's and the widely-used terms "target," "source," "query," and "key."

The rank requirements on $\boldsymbol{W}^Q$ and $\boldsymbol{W}^K$ mean that $\overline{\mathrm{span}}(\{\Phi_{\mathcal{X}}(\boldsymbol{t}), \boldsymbol{t} \in \mathcal{X}\}) = \mathcal{F}_{\mathcal{X}}$ and $\overline{\mathrm{span}}(\{\Phi_{\mathcal{Y}}(\boldsymbol{s}), \boldsymbol{s} \in \mathcal{Y}\}) = \mathcal{F}_{\mathcal{Y}}$. This in turn means that the bilinear mapping is nondegenerate.

*Remark* 3. Now that we have an example of a pair of RKBS's, we can make more concrete some of the discussion from Remark 1. Examining (8a), for example, we see that when we select a $\boldsymbol{k} \in \mathcal{F}_{\mathcal{Y}}$, we define a manifold of functions in $\mathcal{B}_{\mathcal{X}}$ where $\boldsymbol{k}$ is fixed, but $\boldsymbol{W}^Q$ can vary. Similarly, selecting a $\boldsymbol{q} \in \mathcal{F}_{\mathcal{X}}$ defines a manifold in $\mathcal{B}_{\mathcal{Y}}$. Selecting an element from both $\mathcal{F}_{\mathcal{X}}$ and $\mathcal{F}_{\mathcal{Y}}$ locks us into one element each from $\mathcal{B}_{\mathcal{X}}$ and $\mathcal{B}_{\mathcal{Y}}$, which leads to the equality in (6).

*Remark* 4. Examining (8)-(9), we can see that the element drawn from $\mathcal{F}_{\mathcal{Y}}$ that parameterizes the element of $\mathcal{B}_{\mathcal{X}}$, as shown in (8a), is a function of $\Phi_{\mathcal{Y}}$ (and vice-versa for (8b)). This reveals the exact mechanism in which the Transformer-type attention computation is a generalization of the RKBS's considered by Fasshauer et al. (2015), Lin et al. (2019), Xu & Ye (2019), etc., for applications like SVMs, where one of these function spaces is considered fixed.

*Remark* 5. Since the feature maps define the Banach spaces (5), the fact that the parameters $\boldsymbol{W}^Q$ and $\boldsymbol{W}^K$ are learned implies that Transformers learn parametric representations of the RKBS's themselves. This is in contrast to classic kernel methods, where the kernel (and thus the reproducing space) is usually fixed. In fact, in Theorem 2 below, we show that (a variant of) the Transformer architecture can approximate any RKBS mapping.

*Remark* 6. The symmetric version of the exponentiated dot product kernel is known to be a reproducing kernel for the so-called Bargmann space (Bargmann, 1961) which arises in quantum mechanics.

*Remark* 7. Notable in Proposition 1 is that we define the kernel of dot-product attention as including the exponential of the softmax operation. The output of this operation is therefore not the attention scores $a_{ij}$ but rather the unnormalized attention weights, $\bar{\alpha}_{ij} = \alpha_{ij} \sum_j \alpha_{ij}$. Considering the exponential as a part of the kernel operation reveals that the feature spaces for the Transformer are in fact infinite-dimensional in the same sense that the RBF kernel is said to have an infinite-dimensional feature space. In Section 6, we find empirical evidence that this infinite dimensionality may be partially responsible for the Transformer's effectiveness.

## 5 TRANSFORMERS AS KERNEL LEARNERS

### 5.1 THE BINARY RKBS LEARNING PROBLEM AND ITS REPRESENTER THEOREM

Most kernel learning problems take the form of empirical risk minimization problems. For example, if we had a learning problem for a finite dataset $(x_1, z_1), \ldots, (x_n, z_n), x_i \in \mathcal{X}, z_i \in \mathbb{R}$ and wanted to learn a function $f : \mathcal{X} \to \mathbb{R}$ in an RKHS $\mathcal{H}_\kappa$, the learning problem might be written as

$$f^* = \arg\min_{f \in \mathcal{H}_\kappa} \frac{1}{n} \sum_{i=1}^{n} L\left(x_i, z_i, f(x_i)\right) + \lambda R(\|f\|_{\mathcal{H}_\kappa}) \tag{10}$$

where $L : \mathcal{X} \times \mathbb{R} \times \mathbb{R} \to \mathbb{R}$ is a convex loss function, $R : [0, \infty) \to \mathbb{R}$ is a strictly increasing regularization function, and $\lambda$ is a scaling constant. Recent references that consider learning in RKBS's (Georgiev et al., 2014; Fasshauer et al., 2015; Lin et al., 2019; Xu & Ye, 2019) consider similar problems to (10), but with the RKHS $\mathcal{H}$ replaced with an RKBS.

The kernel learning problem for attention, however, is different from (10) in that, as we discussed in the previous section, we need to predict a response $z_{ij}$ (i.e., the attention logit) for every pair $(\boldsymbol{t}_i, \boldsymbol{s}_j)$. This motivates a generalization of the classic class of kernel learning problems that operates on pairs of input spaces. We discuss this generalization now.

**Definition 3** (Binary kernel learning problem - regularized empirical risk minimization). Let $\mathcal{X}$ and $\mathcal{Y}$ be nonempty sets, and $\mathcal{B}_\mathcal{X}$ and $\mathcal{B}_\mathcal{Y}$ RKBS's on $\mathcal{X}$ and $\mathcal{Y}$, respectively. Let $\langle \cdot, \cdot \rangle_{\mathcal{B}_\mathcal{X} \times \mathcal{B}_\mathcal{Y}} : \mathcal{B}_\mathcal{X} \times \mathcal{B}_\mathcal{Y} \to \mathbb{R}$ be a bilinear mapping on the two RKBS's. Let $\Phi_\mathcal{X} : \mathcal{X} \to \mathcal{F}_\mathcal{X}$ and $\Phi_\mathcal{Y} : \mathcal{Y} \to \mathcal{F}_\mathcal{Y}$ be fixed feature mappings with the property that $\langle \Phi_\mathcal{X}(x_i), \Phi_\mathcal{Y}(y_i) \rangle_{\mathcal{F}_\mathcal{X} \times \mathcal{F}_\mathcal{Y}} = \langle f_{\Phi_\mathcal{Y}(y)}, g_{\Phi_\mathcal{X}(x)} \rangle_{\mathcal{B}_\mathcal{X} \times \mathcal{B}_\mathcal{Y}}$. Say $\{x_1, \ldots, x_{n_x}\}, x_i \in \mathcal{X}, \{y_1, \ldots, y_{n_y}\}, y_j \in \mathcal{Y}$, and $\{z_{ij}\}_{i=1,\ldots,n_x; \ j=1,\ldots,n_y}, z_{ij} \in \mathbb{R}$ is a finite dataset where a response $z_{ij}$ is defined for every $(i, j)$ pair of an $x_i$ and a $y_j$. Let $L : \mathcal{X} \times \mathcal{Y} \times \mathbb{R} \times \mathbb{R} \to \mathbb{R}$ be a loss function that is convex for fixed $(x_i, y_j, z_{i,j})$, and $R_\mathcal{X} : [0, \infty) \to \mathbb{R}$ and $R_\mathcal{Y} : [0, \infty) \to \mathbb{R}$ be convex, strictly increasing regularization functions.

A *binary empirical risk minimization kernel learning problem* for learning on a pair of RKBS's takes the form

$$f^*, g^* = \arg\min_{f \in \mathcal{B}_\mathcal{X}, g \in \mathcal{B}_\mathcal{Y}} \left[ \frac{1}{n_x n_y} \sum_{i,j} L\left(x_i, y_j, z_{ij}, \langle f_{\Phi_\mathcal{Y}(y_j)}, g_{\Phi_\mathcal{X}(x_i)} \rangle_{\mathcal{B}_\mathcal{X} \times \mathcal{B}_\mathcal{Y}}\right) \right.$$
$$\left. + \lambda_\mathcal{X} R_\mathcal{X}(\|f\|_{\mathcal{B}_\mathcal{X}}) + \lambda_\mathcal{Y} R_\mathcal{Y}(\|g\|_{\mathcal{B}_\mathcal{Y}}) \right] \tag{11}$$

where $\lambda_\mathcal{X}$ and $\lambda_\mathcal{Y}$ are again scaling constants.

*Remark* 8. The idea of a binary kernel problem that operates over pairs of two sets is not wholly new: there is prior work both in the collaborative filtering (Abernethy et al., 2009) and tensor kernel method (Tao et al., 2005; Kotsia & Patras, 2011; He et al., 2017) literatures. Our problem and results are new in the generalization to Banach rather than Hilbert spaces: as prior work in the RKBS literature (Micchelli et al., 2004; Zhang & Zhang, 2012; Xu & Ye, 2019, etc.) notes, RKBS learning problems are distinct from RKHS ones in their additional nonlinearity and/or nonconvexity. An extension of binary learning problems to Banach spaces is thus motivated by the Transformer setting, where a kernel method is in a context of a nonlinear and nonconvex deep neural network, rather than as a shallow learner like an SVM or matrix completion. For more discussion, see Appendix A.

Virtually all classic kernel learning methods find solutions whose forms are specified by so-called *representer theorems*. Representer theorems state that the solution to a regularized empirical risk minimization problem over a reproducing kernel space can be expressed as a linear combination of evaluations of the reproducing kernel against the dataset. Classic solutions to kernel learning problems thus reduce to finding the coefficients of this linear combination. Representer theorems exist in the literature for RKHS's (Kimeldorf & Wahba, 1971; Schölkopf et al., 2001; Argyriou et al., 2009), RKKS's (Ong et al., 2004; Oglic & Gärtner, 2018), and RKBS's (Zhang et al., 2009; Zhang & Zhang, 2012; Song et al., 2013; Fasshauer et al., 2015; Xu & Ye, 2019; Lin et al., 2019).

Fasshauer et al. (2015, Theorem 3.2), Xu & Ye (2019, Theorem 2.23), and Lin et al. (2019, Theorem 4.7) provide representer theorems for RKBS learning problems. However, their theorems only deal

with learning problems where datapoints come from only one of the sets on which the reproducing kernel is defined (i.e., only $\mathcal{X}$ but not $\mathcal{Y}$), which means the solution sought is an element of only one of the Banach spaces (e.g., $f : \mathcal{X} \to \mathbb{R}, f \in \mathcal{B}_{\mathcal{X}}$). Here, we state and prove a theorem for the more-relevant-to-Transformers binary case presented in Definition 3.

**Theorem 1.** *Suppose we have a kernel learning problem of the form in* (11). *Let* $\kappa : \mathcal{X} \times \mathcal{Y} \to \mathbb{R}$ *be the reproducing kernel of the pair of RKBS's $\mathcal{B}_{\mathcal{X}}$ and $\mathcal{B}_{\mathcal{Y}}$ satisfying Definitions 1 and 2. Then, given some conditions on $\mathcal{B}_{\mathcal{X}}$ and $\mathcal{B}_{\mathcal{Y}}$ (see Appendix B), the regularized empirical risk minimization problem* (11) *has a unique solution pair* $(f^*, g^*)$, *with the property that*

$$\iota(f^*) = \sum_{i=1}^{n_x} \xi_i \kappa(x_i, \cdot) \qquad \iota(g^*) = \sum_{j=1}^{n_y} \zeta_j \kappa(\cdot, y_j). \tag{12}$$

*where $\iota(f)$ (resp. $\iota(g)$) denotes the Gâteaux derivative of the norm of $f$ (resp. $g$) with the convention that $\iota(0) \triangleq 0$, and where $\xi_i, \zeta_j \in \mathbb{R}$.*

*Proof.* See Appendix B. □

### 5.2 A NEW APPROXIMATE KERNEL LEARNING PROBLEM AND UNIVERSAL APPROXIMATION THEOREM

The downside of finding solutions to kernel learning problems like (10) or (11) of the form (12) as suggested by represerter theorems is that they scale poorly to large datasets. It is well-known that for an RKHS learning problem, finding the scalar coefficients by which to multiply the kernel evaluations takes time cubic in the size of the dataset, and querying the model takes linear time. The most popular class of approximation techniques are based on the so-called Nyström method, which constructs a low-rank approximation of the kernel Gram matrix and solves the problem generated by this approximation (Williams & Seeger, 2001). A recent line of work (Gisbrecht & Schleif, 2015; Schleif & Tino, 2017; Oglic & Gärtner, 2019) has extended the Nyström method to RKKS learning.

In this section, we characterize the Transformer learning problem as a new class of approximate kernel methods – a "distillation" approach, one might call it. We formally state this idea now.

**Proposition 2** (Parametric approximate solutions of binary kernel learning problems)**.** *Consider the setup of a binary kernel learning problem from Definition 3. We want to find approximations to the solution pair $(f^*, g^*)$. In particular, we will say we want an approximation $\hat{\kappa} : \mathcal{X} \times \mathcal{Y} \to \mathbb{R}$ such that*

$$\hat{\kappa}(x, y) \approx \left\langle f^*_{\Phi_{\mathcal{Y}}(y)}, g^*_{\Phi_{\mathcal{X}}(x)} \right\rangle_{\mathcal{B}_{\mathcal{X}} \times \mathcal{B}_{\mathcal{Y}}} \qquad \text{for all } x \in \mathcal{X} \text{ and } y \in \mathcal{Y}. \tag{13}$$

*Comparing* (13) *to* (6) *suggests a solution: to learn a function $\hat{\kappa}$ that approximates $\kappa$. In particular,* (6) *suggests learning explicit approximations of the feature maps, i.e.,*

$$\hat{\kappa}(x, y) \approx \langle \Phi_{\mathcal{X}}(x), \Phi_{\mathcal{Y}}(y) \rangle_{\mathcal{F}_{\mathcal{X}} \times \mathcal{F}_{\mathcal{Y}}} .$$

*In fact, it turns out that the Transformer query-key mapping* (1) *does exactly this. That is, while the Transformer kernel calculation outlined in Propostion 1 is finite-dimensional, it can in fact approximate the potentially infinite-dimensional optimal solution $(f^*, g^*)$ characterized in Theorem 1. This fact is proved next.*

**Theorem 2.** *Let $\mathcal{X} \subset \mathbb{R}^{d_t}$ and $\mathcal{Y} \subset \mathbb{R}^{d_s}$ be compact; $\boldsymbol{t} \in \mathcal{X}, \boldsymbol{s} \in \mathcal{Y}$; and let $q_\ell : \mathcal{X} \to \mathbb{R}$ and $k_\ell : \mathcal{Y} \to \mathbb{R}$ for $\ell = 1, \ldots, d$ be two-layer neural networks with $m$ hidden units. Then, for any continuous function $F : \mathcal{X} \times \mathcal{Y} \to \mathbb{R}$ and $\epsilon > 0$, there are integers $m, d > 0$ such that*

$$\left| F(\boldsymbol{t}, \boldsymbol{s}) - \sum_{\ell=1}^{d} q_\ell(\boldsymbol{t}) k_\ell(\boldsymbol{s}) \right| < \epsilon \quad \text{for all } \boldsymbol{t} \in \mathcal{X}, \boldsymbol{s} \in \mathcal{Y}. \tag{14}$$

*Proof.* See Appendix C. □

We now outline how Theorem 2 relates to Transformers. If we concatenate the outputs of the two-layer neural networks $\{q_\ell\}_{\ell=1}^{d}$ and $\{k_\ell\}_{\ell=1}^{d}$ into $d$-dimensional vectors $\boldsymbol{q} : \mathbb{R}^{d_t} \to \mathbb{R}^d$ and $\boldsymbol{k} : \mathbb{R}^{d_s} \to \mathbb{R}^d$, then the dot product $\boldsymbol{q}(\boldsymbol{t})^\mathsf{T} \boldsymbol{k}(\boldsymbol{s})$ denoted by the sum in (14) can approximate any

real-valued continuous function on $\mathcal{X} \times \mathcal{Y}$. Minus the usual caveats in applications of universal approximation theorems (i.e., in practice the output elements share hidden units rather than having independent ones), this dot product is exactly the computation of the attention logits $a_{ij}$, i.e., $F(\boldsymbol{t}, \boldsymbol{s}) \approx \log \kappa(\boldsymbol{t}, \boldsymbol{s})$ for the $F$ in (14) and the $\kappa$ in (9) up to a scaling constant $\sqrt{d}$.

Since the exponential mapping between the attention logits and the exponentiated query-key kernel used in Transformers is a one-to-one mapping, if we take $F(\boldsymbol{t}, \boldsymbol{s}) = \log \left\langle f^*_{\Phi_\mathcal{Y}(\boldsymbol{s})}, g^*_{\Phi_\mathcal{X}(\boldsymbol{t})} \right\rangle_{\mathcal{B}_\mathcal{X} \times \mathcal{B}_\mathcal{Y}}$, then we can use a Transformer's dot-product attention to approximate the optimal solution to any RKBS solution arbitrarily well.

The core idea of an attention-based deep neural network is then to learn parametric representations of $q_\ell$ and $k_\ell$ via stochastic gradient descent. Unlike traditional represener-theorem-based learned functions, training time of attention-based kernel machines like deep Transformers (generally, but with no guarantees) scale sub-cubically with dataset size, and evaluation time stays constant regardless of dataset size.

## 6    IS THE EXPONENTIATED DOT PRODUCT ESSENTIAL TO TRANSFORMERS?

Table 1: Test BLEU scores for Transformers with various kernels on machine translation (case-sensitive sacreBLEU). Values are mean $\pm$ std. dev over 5 training runs with different random seeds.

|  | EDP | RBF | L2 Distance | EI | Quadratic |
|---|---|---|---|---|---|
| IWSLT14 DE-EN | $30.41 \pm 0.03$ | $30.32 \pm 0.22$ | $19.45 \pm 0.16$ | $30.84 \pm 0.27$ | $29.56 \pm 0.19$ |
| WMT14 EN-FR | $35.11 \pm 0.08$ | $35.57 \pm 0.20$ | $28.41 \pm 0.26$ | $34.51 \pm 0.17$ | $34.54 \pm 0.30$ |

EDP = Exponentiated dot product; EI = Exponentiated intersection kernel

We study modifications of the Transformer with several kernels used in classic kernel machines. We train on two standard machine translation datasets and two standard sentiment classification tasks. For machine translation, IWSLT14 DE-EN is a relatively small dataset, while WMT14 EN-FR is a considerably larger one. For sentiment classification, we consider SST-2 and SST-5. We retain the standard asymmetric query and key feature mappings, i.e., $\boldsymbol{q} = \boldsymbol{W}^Q \boldsymbol{t}$ and $\boldsymbol{k} = \boldsymbol{W}^K \boldsymbol{s}$, and only modify the kernel $\kappa : \mathbb{R}^d \times \mathbb{R}^d \to \mathbb{R}_{\geq 0}$. In the below, $\tau > 0$ and $\gamma \in \mathbb{R}$ are per-head learned scalars.

Our kernels of interest are:

1. the (scaled) exponentiated dot product (EDP), $\kappa(\boldsymbol{q}, \boldsymbol{t}) = \exp(\boldsymbol{q}^T \boldsymbol{k} / \sqrt{d})$, i.e., the standard Transformer kernel;

2. the radial basis function (RBF) kernel, $\kappa(\boldsymbol{q}, \boldsymbol{t}) = \exp(\| -\tau/\sqrt{d}(\boldsymbol{q} - \boldsymbol{k}) \|_2^2)$, where $\| \cdot \|_2$ is the standard 2-norm. It is well-known that the RBF kernel is a normalized version of the exponentiated dot-product, with the normalization making it translation-invariant;

3. the vanilla L2 distance, $\kappa(\boldsymbol{q}, \boldsymbol{t}) = \tau/\sqrt{d} \| \boldsymbol{q} - \boldsymbol{k} \|_2$;

4. an exponentiated version of the intersection kernel, $\kappa(\boldsymbol{q}, \boldsymbol{t}) = \exp(\sum_{\ell=1}^{d} \min(q_\ell, k_\ell))$. The symmetric version of the intersection kernel was popular in kernel machines for computer vision applications (Barla et al., 2003; Grauman & Darrell, 2005; Maji et al., 2008, etc.), and is usually characterized as having an associated RKHS that is a subspace of the function space $L^2$ (i.e., it is infinite-dimensional in the sense of having a feature space of continuous functions, as opposed to the infinite-dimensional infinite series of the EDP and RBF kernels);

5. a quadratic polynomial kernel, $\kappa(\boldsymbol{q}, \boldsymbol{k}) = (1/\sqrt{d} \boldsymbol{q}^\mathsf{T} \boldsymbol{k} + \gamma)^2$.

Full implementation details are provided in Appendix D.

Results for machine translation are presented in Table 1. Several results stand out. First, the exponentiated dot product, RBF, and exponentiated intersection kernels, which are said to have infinite-dimensional feature spaces, indeed do perform better than kernels with lower-dimensional feature maps such as the quadratic kernel. In fact, the RBF and EDP kernels perform about the same, suggesting that a deep Transformer may not need the translation-invariance that makes the RBF

Table 2: Test accuracies for Transformers with various kernels on sentiment classification. Values are mean ± std. dev over 5 training runs with different random seeds.

|            | EDP           | RBF           | L2 Distance   | EI            | Quadratic     |
|------------|---------------|---------------|---------------|---------------|---------------|
| SST-2 (%)  | $76.70 \pm 0.36$ | $74.24 \pm 0.39$ | $76.78 \pm 0.67$ | $74.90 \pm 1.32$ | $76.24 \pm 0.65$ |
| SST-5 (%)  | $39.44 \pm 0.47$ | $39.04 \pm 0.62$ | $39.44 \pm 1.33$ | $37.74 \pm 0.48$ | $39.34 \pm 0.80$ |

EDP = Exponentiated dot product; EI = Exponentiated intersection kernel

kernel preferred to the EDP in classic kernel machines. Intriguingly, the (unorthodox) exponentiated intersection kernel performs about the same as the two than the EDP and RBF kernels on IWSLT14 DE-EN, but slightly worse on WMT14 EN-FR. As mentioned, the EDP and RBF kernels have feature spaces of infinite series, while the intersection kernel corresponds to a feature space of continuous functions. On both datasets, the quadratic kernel performs slightly worse than the best infinite-dimensional kernel, while the L2 distance performs significantly worse.

Results for sentiment classification appear in Table 2. Unlike the machine translation experiments, the infinite-dimensional kernels do not appear strictly superior to the finite-dimensional ones on this task. In fact, the apparent loser here is the exponentiated intersection kernel, while the L2 distance, which performed the worst on machine translation, is within a standard deviation of the top-performing kernel. Notably, however, the variance of test accuracies on sentiment classification means that it is impossible to select a statistically significant "best" on this task. It is possible that the small inter-kernel variation relates to the relative simplicity of this problem (and relative smallness of the dataset) vs. machine translation: perhaps an infinite-dimensional feature space is not needed to obtain Transformer-level performance on this learning problem.

It is worth noting that the exponentiated dot product kernel (again, the standard Transformer kernel) is a consistent high performer. This may be experimental evidence for the practical usefulness of the universal approximation property they enjoy (c.f. Theorem 2).

The relatively small yet statistically significant performance differences between kernels is reminiscent of the same phenomenon with activation functions (ReLU, ELU, etc.) for neural nets. Moreover, the wide inter-kernel differences in performance for machine translation, compared against the much smaller performance differences on the SST sentiment analysis tasks, demonstrates an opportunity for future study on this apparent task- and dataset-dependency. As a whole, these results suggest that kernel choice may be an additional design parameter for Transformer networks.

## 7 CONCLUSION

In this paper, we drew connections between classic kernel methods and the state-of-the-art Transformer networks. Beyond the theoretical interest in developing new RKBS representer theorems and other kernel theory, we gained new insight into what may make Transformers work. Our experimental results suggest that the infinite dimensionality of the Transformer kernel makes it a good choice in application, similar to how the RBF kernel is the standard choice for e.g. SVMs. Our work also reveals new avenues for Transformer research. For example, our experimental results suggest that choice of Transformer kernel acts as a similar design choice as activation functions in neural net design. Among the new open research questions are (1) whether the exponentiated dot-product should be always preferred, or if different kernels are better for different tasks (c.f. how GELUs have recently become very popular as replacements for ReLUs in Transformers), (2) any relation between vector-valued kernels used for structured prediction (Álvarez et al., 2012) and, e.g., multiple attention heads, and (3) the extension of Transformer-type deep kernel learners to non-Euclidean data (using, e.g., graph kernels or kernels on manifolds).

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

# A  DEEP NEURAL NETWORKS LEAD TO BANACH SPACE ANALYSIS

Examining the kernel learning problem (11), it may not immediately clear why the reproducing spaces on $\mathcal{X}$ and $\mathcal{Y}$ need be Banach spaces rather than Hilbert spaces. Suppose for example that we have two RKHS's $\mathcal{H}_{\mathcal{X}}$ and $\mathcal{H}_{\mathcal{Y}}$ on $\mathcal{X}$ and $\mathcal{Y}$, respectively. Then, we can take their tensor product $\mathcal{H}_{\mathcal{X}} \otimes \mathcal{H}_{\mathcal{Y}}$ as an RKHS on $\mathcal{X} \times \mathcal{Y}$, with associated reproducing kernel $\kappa_{\mathcal{X} \times \mathcal{Y}}(x_1 \otimes y_1, x_2 \otimes y_2) = \kappa_{\mathcal{X}}(x_1, x_2)\kappa_{\mathcal{Y}}(y_1, y_2)$, where $\kappa_{\mathcal{X}}$ and $\kappa_{\mathcal{Y}}$ are the reproducing kernels of $\mathcal{H}_{\mathcal{X}}$ and $\mathcal{H}_{\mathcal{Y}}$, respectively, and $x_1 \otimes y_1, x_2 \otimes y_2 \in \mathcal{X} \otimes \mathcal{Y}$. The solutions to a regularized kernel problem like (11) would then be drawn from $\mathcal{H}_{\mathcal{X}}$ and $\mathcal{H}_{\mathcal{Y}}$. This setup is similar to those studied in, e.g., Abernethy et al. (2009); He et al. (2017).

In a shallow kernel learner like an SVM, the function in the RKHS can be characterized via its norm. Representer theorems allow for the norm of the function in the Hilbert space to be calculated from the scalar coefficients that make up the solution. On the other hand, for a Transformer layer in a multilayer neural network, regularization is usually not done via norm penalty as shown in (11). In most applications, regularization is done via dropout on the attention weights $a_{ij}$ as well as via the implicit regularization obtained from subsampling the dataset during iterations of stochastic gradient descent. While dropout has been characterized as a form of weight decay (i.e., a variant of p-norm penalization) for linear models (Baldi & Sadowski, 2013; Wager et al., 2013, etc.), recent work has shown that dropout induces a more complex regularization effect in deep networks (Helmbold & Long, 2017; Arora et al., 2020, etc.). Thus, it is difficult to characterize the norm of the vector spaces we are traversing when solving the general problem (11) in the context of a deep network. This can lead to ambiguity as to whether the norm being regularized as we traverse the solution space is a Hilbert space norm. If $f$ and $g$ are infinite series or $L^p$ functions, for example, their resident space is only a Hilbert space if the associated norm is the $\ell^2$ or $L^2$ norm. This motivates the generalization of kernel learning theorems to the general Banach space setting when in the context of deep neural networks.

# B  PROOF OF THEOREM 1

## B.1  PRELIMINIARIES

To prove this theorem, we first need some results and definitions regarding various properties of Banach spaces (Megginson, 1998). These preliminaries draw from Xu & Ye (2019) and Lin et al. (2019).

Two metric spaces $(\mathcal{M}, d_{\mathcal{M}})$ and $(\mathcal{N}, d_{\mathcal{N}})$ are said to be **isometrically isomorphic** if there exists a bijective mapping $T : \mathcal{M} \to \mathcal{N}$, called an isometric isomorphism, such that for all $m \in \mathcal{M}$, $d_{\mathcal{N}}(T(m)) = d_{\mathcal{M}}(m)$ (Megginson, 1998, Definition 1.4.13).

The **dual space** of a vector space $\mathcal{V}$ over a field $\mathbb{F}$, which we will denote $\mathcal{V}^*$, is the space of all continuous linear functionals on $\mathcal{V}$, i.e.,

$$\mathcal{V}^* = \{g : \mathcal{V} \to \mathbb{F}, g \text{ linear and continuous}\}. \tag{B.1}$$

A normed vector space $\mathcal{V}$ is **reflexive** if it is isometrically isomorphic to $\mathcal{V}^{**}$, the dual space of its dual space (a.k.a. its double dual).

For a normed vector space $\mathcal{V}$, the **dual bilinear product**, which we will denote $\langle \cdot, \cdot \rangle_{\mathcal{V}}$ (i.e., with only one subscript, as opposed to e.g., $\langle \cdot, \cdot \rangle_{\mathcal{B}_{\mathcal{X}} \times \mathcal{B}_{\mathcal{Y}}}$), is defined on $\mathcal{V}$ and $\mathcal{V}^*$ as

$$\langle f, g \rangle_{\mathcal{V}} \triangleq g(f) \quad \text{for } f \in \mathcal{V}, g \in \mathcal{V}^*.$$

Given a normed vector space $\mathcal{V}$ and its dual space $\mathcal{V}^*$, let $\mathcal{U} \subseteq \mathcal{V}$ and $\mathcal{W} \subseteq \mathcal{V}^*$. The **annihilator** of $\mathcal{U}$ in $\mathcal{V}^*$ and the **annihilator** of $\mathcal{W}$ in $\mathcal{V}$, denoted $\mathcal{U}^{\perp}$ and $^{\perp}\mathcal{W}$ respectively, are (Megginson, 1998, Definition 1.10.14)

$$\mathcal{U}^{\perp} = \{g \in \mathcal{V}^* : \langle f, g \rangle_{\mathcal{V}} = 0 \quad \forall f \in \mathcal{U}\}$$
$$^{\perp}\mathcal{W} = \{f \in \mathcal{V} : \langle f, g \rangle_{\mathcal{V}} = 0 \quad \forall g \in \mathcal{W}\}.$$

A normed vector space $\mathcal{V}$ is called **strictly convex** if $\|tv_1 + (1-t)v_2\|_{\mathcal{V}} < 1$ whenever $\|v_1\|_{\mathcal{V}} = \|v_2\|_{\mathcal{V}} = 1$, $0 < t < 1$, where $v_1, v_2 \in \mathcal{V}$ and $\| \cdot \|_{\mathcal{V}}$ denotes the norm of $\mathcal{V}$ (Megginson, 1998, Definition 5.1.1; citing Clarkson, 1936 and Akhiezer & Krein, 1962).

A nonempty subset $\mathcal{A}$ of a metric space $(\mathcal{M}, d_{\mathcal{M}})$ is called a **Chebyshev set** if, for every element $m \in \mathcal{M}$, there is exactly one element $c \in \mathcal{A}$ such that $d_{\mathcal{M}}(m,c) = d_{\mathcal{M}}(m, \mathcal{A})$ (Megginson, 1998, Definition 5.1.17) (where recall the distance between a point $m$ and a set $\mathcal{A}$ in a metric space is equal to $\inf_{c \in \mathcal{A}} d_{\mathcal{M}}(m,c)$). If a normed vector space $\mathcal{V}$ is reflexive and strictly convex, then every nonempty closed convex subset of $\mathcal{V}$ is a Chebyshev set (Megginson, 1998, Corollary 5.1.19; citing Day, 1941).

For a normed vector space $\mathcal{V}$ and $v, w \in \mathcal{V}$, the **Gâteaux derivative of the norm** (Megginson, 1998, Definition 5.4.15) at $v$ in the direction of $w$ is defined as

$$\lim_{t \to 0} \frac{\|v + tw\|_{\mathcal{V}} - \|v\|_{\mathcal{V}}}{t}.$$

If the Gâteaux derivative of the norm at $v$ in the direction of $w$ exists for all $w \in \mathcal{V}$, then $\| \cdot \|_{\mathcal{V}}$ is said to be Gâteaux differentiable at $v$. A normed vector space $\mathcal{V}$ is called Gâteaux differentiable or **smooth** if its norm is Gâteaux differentiable at all $v \in \mathcal{V}$ (Megginson, 1998, Corollary 5.4.18).

The smoothness of a normed vector space $\mathcal{V}$ implies that, if we define a "norm operator" $\rho$ on $\mathcal{V}$, $\rho(v) \triangleq \|v\|_{\mathcal{V}}$, then for each $v \in \mathcal{V} \setminus \{0\}$, there exists a continuous linear functional $d_G\rho(v)$ on $\mathcal{V}$ such that (Xu & Ye, 2019, p. 24)

$$\langle w, d_G\rho(v) \rangle_{\mathcal{V}} = \lim_{t \to 0} \frac{\|v + tw\|_{\mathcal{V}} - \|v\|_{\mathcal{V}}}{t} \text{ for all } w \in \mathcal{V}.$$

Since the Gâteaux derivative of the norm is undefined at 0, following Xu & Ye (2019, Equation 2.16); Lin et al. (2019, p. 20); etc., we define a regularized Gâteaux derivative of the norm operator on $\mathcal{V}$,

$$\iota(v) \triangleq \begin{cases} d_G\rho(v) & \text{when } v \neq 0 \\ 0 & \text{when } v = 0 \end{cases} \tag{B.2}$$

for $v \in \mathcal{V}$.

Given two vector spaces $\mathcal{V}$ and $\mathcal{W}$ defined over a field $\mathbb{F}$, the **direct sum**, denoted $\mathcal{V} \oplus \mathcal{W}$, is the vector space with elements $(v, w) \in \mathcal{V} \oplus \mathcal{W}$ for $v \in \mathcal{V}$, $w \in \mathcal{W}$ with the additional structure

$$(v_1, w_1) + (v_2, w_2) = (v_1 + v_2, w_1 + w_2) \quad \text{for } v_1, v_2 \in \mathcal{V}, w_1, w_2 \in \mathcal{W}$$
$$c(v, w) = (cv, cw) \quad \text{for } v \in \mathcal{V}, w \in \mathcal{W}, c \in \mathbb{F}.$$

If $\mathcal{V}$ and $\mathcal{W}$ are normed vector spaces with norms $\| \cdot \|_{\mathcal{V}}$ and $\| \cdot \|_{\mathcal{W}}$, respectively, then we will say $\mathcal{V} \oplus \mathcal{W}$ has the norm

$$\|v, w\|_{\mathcal{V} \oplus \mathcal{W}} \triangleq \|v\|_{\mathcal{V}} + \|w\|_{\mathcal{W}} \text{ for } v \in \mathcal{V}, w \in \mathcal{W}. \tag{B.3}$$

Megginson (1998, Definition 1.8.1) calls (B.3) a "1-norm" direct sum norm, but notes that other norm-equivalent direct sum norms such as a 2-norm and infinity-norm are possible. Some other useful facts about direct sums are:

- if $\mathcal{V}$ and $\mathcal{W}$ are both strictly convex, then $\mathcal{V} \oplus \mathcal{W}$ is strictly convex (Megginson, 1998, Theorem 5.1.23);
- if $\mathcal{V}$ and $\mathcal{W}$ are both reflexive, then $\mathcal{V} \oplus \mathcal{W}$ is reflexive (Megginson, 1998, Corollary 1.11.20);
- and if $\mathcal{V}$ and $\mathcal{W}$ are both smooth, then $\mathcal{V} \oplus \mathcal{W}$ is smooth (Megginson, 1998, Theorem 5.4.22).

An element $v$ of a normed vector space $\mathcal{V}$ is said to be **orthogonal** (or Birkhoff-James orthogonal) to another element $w \in \mathcal{V}$ if (Birkhoff, 1935; James, 1947)

$$\|v + tw\|_{\mathcal{V}} \geq \|v\|_{\mathcal{V}} \text{ for all } t \in \mathbb{R}.$$

If $\mathcal{W} \subseteq \mathcal{V}$, then we say $v \in \mathcal{V}$ is orthogonal to $\mathcal{W}$ if $v$ is orthogonal to all $w \in \mathcal{W}$.

Now we can state a lemma regarding orthogonality in RKBS's.

**Lemma 1** (Xu & Ye, 2019, Lemma 2.21). *If the RKBS $\mathcal{B}$ is smooth, then $f \in \mathcal{B}$ is orthogonal to $g \in \mathcal{B}$ if and only if $\langle g, \iota(f) \rangle_{\mathcal{B}} = 0$, where $\langle \cdot, \cdot \rangle_{\mathcal{B}}$ means the dual bilinear product as given in (B.1) and $\iota$ is the regularized Gâteaux derivative from (B.2). Also, an $f \in \mathcal{B} \setminus \{0\}$ is orthogonal to a subspace $\mathcal{N} \subseteq \mathcal{B}$ if and only if $\langle h, \iota(f) \rangle_{\mathcal{B}} = 0$ for all $h \in \mathcal{N}$.*

## B.2 MINIMUM-NORM INTERPOLATION (OPTIMAL RECOVERY)

Following Fasshauer et al. (2015); Xu & Ye (2019); Lin et al. (2019), we first prove a representer theorem for a simpler problem – that of perfect interpolation while minimizing the norm of the solution – before proceeding to the representer theorem for the empirical risk minimization problem (11) in the next section.

**Definition 4** (Minimum-norm interpolation in a pair of RKBS's). Let $\mathcal{X}$ and $\mathcal{Y}$ be nonempty sets, and $\mathcal{B}_\mathcal{X}$ and $\mathcal{B}_\mathcal{Y}$ RKBS's on $\mathcal{X}$ and $\mathcal{Y}$, respectively. Let $\langle \cdot, \cdot \rangle_{\mathcal{B}_\mathcal{X} \times \mathcal{B}_\mathcal{Y}} : \mathcal{B}_\mathcal{X} \times \mathcal{B}_\mathcal{Y} \to \mathbb{R}$ be a bilinear mapping on the two RKBS's, $\Phi_\mathcal{X} : \mathcal{X} \to \mathcal{F}_\mathcal{X}$ and $\Phi_\mathcal{Y} : \mathcal{Y} \to \mathcal{F}_\mathcal{Y}$. Let $\kappa(x_i, y_i) = \langle \Phi_\mathcal{X}(x), \Phi_\mathcal{Y}(y) \rangle_\mathcal{F} = \langle f_{\Phi_\mathcal{Y}(y_i)}, g_{\Phi_\mathcal{X}(x_i)} \rangle_{\mathcal{B}_\mathcal{X} \times \mathcal{B}_\mathcal{Y}}$ be a reproducing kernel of $\mathcal{X}$ and $\mathcal{Y}$ satisfying Definitions 1 and 2. Say $\{x_1, \ldots, x_{n_x}\}, x_i \in \mathcal{X}, \{y_1, \ldots, y_{n_y}\}, y_i \in \mathcal{Y}$, and $\{z_{ij}\}_{i=1,\ldots,n_x;\ j=1,\ldots,n_y}, z_{ij} \in \mathbb{R}$ is a finite dataset where a response $z_{ij}$ is defined for every $(i, j)$ pair of an $x_i$ and a $y_j$. The minimum-norm interpolation problem is

$$f^*, g^* = \operatorname*{arg\,min}_{f \in \mathcal{B}_\mathcal{X}, y \in \mathcal{B}_\mathcal{Y}} \|f\|_{\mathcal{B}_\mathcal{X}} + \|g\|_{\mathcal{B}_\mathcal{Y}}$$
$$\text{such that } (f, g) \in \mathcal{N}_{\mathcal{X}, \mathcal{Y}, \mathcal{Z}} \tag{B.4}$$

where

$$\mathcal{N}_{\mathcal{X}, \mathcal{Y}, \mathcal{Z}} = \{(f, g) \in \mathcal{B}_\mathcal{X} \oplus \mathcal{B}_\mathcal{Y} \text{ s.t. } \langle f_{\Phi_\mathcal{Y}(y_j)}, g_{\Phi_\mathcal{X}(x_i)} \rangle_{\mathcal{B}_\mathcal{X} \times \mathcal{B}_\mathcal{Y}} = z_{ij}\ \forall\, i, j\}. \tag{B.5}$$

To discuss the solution of (B.4), we first need to establish the condition for the existence of a solution. The following is a generalization of a result from Section 2.6 of Xu & Ye (2019).

**Lemma 2.** *If the set $\{\kappa(x_i, \cdot)\}_{i=1}^{n_x}$ is linearly independent in $\mathcal{B}_\mathcal{Y}$, the set $\{\kappa(\cdot, y_j)\}_{j=1}^{n_y}$ is linearly independent in $\mathcal{B}_\mathcal{X}$, and the bilinear mapping $\langle \cdot, \cdot \rangle_{\mathcal{B}_\mathcal{X} \times \mathcal{B}_\mathcal{Y}} : \mathcal{B}_\mathcal{X} \times \mathcal{B}_\mathcal{Y} \to \mathbb{R}$ is nondegenerate, then $\mathcal{N}_{\mathcal{X}, \mathcal{Y}, \mathcal{Z}}$ (B.5) is nonempty.*

*Proof.* From the definition of $\kappa$ (3) and the bilinearity of $\langle \cdot, \cdot \rangle_{\mathcal{B}_\mathcal{X} \times \mathcal{B}_\mathcal{Y}}$, we can write that

$$\left\langle f, \sum_{i=1}^{n_x} c_i \kappa(x_i, \cdot) \right\rangle_{\mathcal{B}_\mathcal{X} \times \mathcal{B}_\mathcal{Y}} = \sum_{i=1}^{n_x} c_i \langle f, \kappa(x_i, \cdot) \rangle_{\mathcal{B}_\mathcal{X} \times \mathcal{B}_\mathcal{Y}} = \sum_{i=1}^{n_x} c_i f(x_i) \text{ for all } f \in \mathcal{B}_\mathcal{X}$$

for $c_i \in \mathbb{R}$, and that

$$\left\langle \sum_{j=1}^{n_y} c_j \kappa(\cdot, y_j), g \right\rangle_{\mathcal{B}_\mathcal{X} \times \mathcal{B}_\mathcal{Y}} = \sum_{j=1}^{n_y} c_j \langle \kappa(\cdot, y_j), g \rangle_{\mathcal{B}_\mathcal{X} \times \mathcal{B}_\mathcal{Y}} = \sum_{j=1}^{n_y} c_j g(y_j) \text{ for all } g \in \mathcal{B}_\mathcal{Y}$$

for $c_j \in \mathbb{R}$. This means that

$$\sum_{i=1}^{n_x} c_i \kappa(x_i, \cdot) = 0 \text{ if and only if } \sum_{i=1}^{n_x} c_i f(x_i) = 0 \text{ for all } f \in \mathcal{B}_\mathcal{X}$$

and

$$\sum_{j=1}^{n_y} c_j \kappa(\cdot, y_j) = 0 \text{ if and only if } \sum_{j=1}^{n_y} c_j g(y_j) = 0 \text{ for all } g \in \mathcal{B}_\mathcal{Y}.$$

This shows that linear independence of $\{\kappa(x_i, \cdot)\}_{i=1}^{n_x}$ and $\{\kappa(\cdot, y_j)\}_{j=1}^{n_y}$ imply linear independence of $\{f(x_i)\}_{i=1}^{n_x}$ and $\{g(y_j)\}_{j=1}^{n_y}$, respectively. Then, considering the nondegeneracy of the bilinear mapping $\langle \cdot, \cdot \rangle_{\mathcal{B}_\mathcal{X} \times \mathcal{B}_\mathcal{Y}}$, we can say that

$$\left\langle \sum_{j=1}^{n_y} c_j \kappa(\cdot, y_j), \sum_{i=1}^{n_x} c_i \kappa(x_i, \cdot) \right\rangle_{\mathcal{B}_\mathcal{X} \times \mathcal{B}_\mathcal{Y}} = 0$$

$$\text{if and only if } \left( \sum_{i=1}^{n_x} c_i f(x_i) = 0 \text{ for all } f \in \mathcal{B}_\mathcal{X} \quad \text{or} \quad \sum_{j=1}^{n_y} c_j g(y_j) = 0 \text{ for all } g \in \mathcal{B}_\mathcal{Y} \right).$$

From this, we can see that linear independence of $\{\kappa(x_i, \cdot)\}_{i=1}^{n_x}$ and $\{\kappa(\cdot, y_j)\}_{j=1}^{n_y}$, and the nondegeneracy of $\langle \cdot, \cdot \rangle_{\mathcal{B}_\mathcal{X} \times \mathcal{B}_\mathcal{Y}}$ ensure the existence of at least one $(f^*, g^*)$ pair in $\mathcal{N}_{\mathcal{X}, \mathcal{Y}, \mathcal{Z}}$. $\square$

Now we can prove a lemma characterizing the solution to (B.4).

**Lemma 3.** *Consider the minimum-norm interpolation problem from Definition 4. Assume that $\mathcal{B}_\mathcal{X}$ and $\mathcal{B}_\mathcal{Y}$ are smooth, strictly convex, and reflexive,[2] and that $\{\kappa(x_i, \cdot)\}_{i=1}^{n_x}$ and $\{\kappa(\cdot, y_j)\}_{j=1}^{n_y}$ are linearly independent. Then, (B.4) has a unique solution pair $(f^*, g^*)$, with the property that*

$$\iota(f^*) = \sum_{i=1}^{n_x} \xi_i \kappa(x_i, \cdot) \qquad \iota(g^*) = \sum_{j=1}^{n_y} \zeta_j \kappa(\cdot, y_j).$$

*where $\iota(\cdot)$ is the regularized Gâteaux derivative as defined in* (B.2)*, and where $\xi_i, \zeta_j \in \mathbb{R}$.*

*Proof.* The existence of a solution pair is given by the linear independence of $\{\kappa(x_i, \cdot)\}_{i=1}^{n_x}$ and $\{\kappa(\cdot, y_j)\}_{j=1}^{n_y}$ and Lemma 2. Uniqueness of the solution will be shown by showing that $\mathcal{N}_{\mathcal{X},\mathcal{Y},\mathcal{Z}}$ is closed and convex and a subset of a strictly convex and reflexive set, which ensures it is a Chebyshev set.

Since $\mathcal{B}_\mathcal{X}$ and $\mathcal{B}_\mathcal{Y}$ are strictly convex and reflexive, their direct sum $\mathcal{B}_\mathcal{X} \oplus \mathcal{B}_\mathcal{Y}$ is strictly convex and reflexive, as we noted in section B.1.

Now we analyze $\mathcal{N}_{\mathcal{X},\mathcal{Y},\mathcal{Z}}$. We first show convexity. Pick any $(f, g), (f', g') \in \mathcal{N}_{\mathcal{X},\mathcal{Y},\mathcal{Z}}$ and $t \in (0, 1)$. Then note that for any $(x_i, y_j, z_{i,j})$,

$$\left\langle t f_{\Phi_\mathcal{Y}(y_j)}, t g_{\Phi_\mathcal{X}(x_i)} \right\rangle_{\mathcal{B}_\mathcal{X} \times \mathcal{B}_\mathcal{Y}} + \left\langle (1-t) f'_{\Phi_\mathcal{Y}(y_j)}, (1-t) g'_{\Phi_\mathcal{X}(x_i)} \right\rangle_{\mathcal{B}_\mathcal{X} \times \mathcal{B}_\mathcal{Y}}$$

$$= t \left\langle f_{\Phi_\mathcal{Y}(y_j)}, g_{\Phi_\mathcal{X}(x_i)} \right\rangle_{\mathcal{B}_\mathcal{X} \times \mathcal{B}_\mathcal{Y}} + (1-t) \left\langle f'_{\Phi_\mathcal{Y}(y_j)}, g'_{\Phi_\mathcal{X}(x_i)} \right\rangle_{\mathcal{B}_\mathcal{X} \times \mathcal{B}_\mathcal{Y}}$$

$$= t z_{i,j} + (1-t) z_{i,j}$$

$$= z_{i,j}$$

thus showing that $\mathcal{N}_{\mathcal{X},\mathcal{Y},\mathcal{Z}}$ is convex. Closedness may be shown by the strict convexity of its superset $\mathcal{B}_\mathcal{X} \oplus \mathcal{B}_\mathcal{Y}$ and the continuity of $\langle \cdot, \cdot \rangle_{\mathcal{B}_1 \times \mathcal{B}_2}$. Thus, the closed and convex $\mathcal{N}_{\mathcal{X},\mathcal{Y},\mathcal{Z}} \subseteq \mathcal{B}_\mathcal{X} \oplus \mathcal{B}_\mathcal{Y}$ is a Chebyshev set, implying a unique $(f^*, g^*) \in \mathcal{N}_{\mathcal{X},\mathcal{Y},\mathcal{Z}}$ with

$$\|f^*, g^*\|_{\mathcal{B}_\mathcal{X} \oplus \mathcal{B}_\mathcal{Y}} = \min_{(f,g) \in \mathcal{N}_{\mathcal{X},\mathcal{Y},\mathcal{Z}}} \|f\|_{\mathcal{B}_\mathcal{X}} + \|g\|_{\mathcal{B}_\mathcal{Y}}.$$

Now we characterize this solution $(f^*, g^*)$. Similar to proofs of the classic RKHS representer theorem (Schölkopf et al., 2001) and those of earlier RKBS representer theorems (Xu & Ye, 2019; Lin et al., 2019, etc.), we approach this via orthogonal decomposition. Consider the following set of function pairs $(f, g)$ that map all data pairs $(x_i, y_j)$ to 0:

$$\mathcal{N}_{\mathcal{X},\mathcal{Y},0} = \Big\{ (f, g) \in \mathcal{B}_\mathcal{X} \oplus \mathcal{B}_\mathcal{Y} : \quad \left\langle f_{\Phi_\mathcal{Y}(y_j)}, g_{\Phi_\mathcal{X}(x_i)} \right\rangle_{\mathcal{B}_\mathcal{X} \times \mathcal{B}_\mathcal{Y}} = 0;$$

$$i = 1, \dots, n_x; \quad j = 1, \dots, n_y \Big\}$$

We can see that $\mathcal{N}_{\mathcal{X},\mathcal{Y},0}$ is closed under addition and scalar multiplication, making it a subspace of $\mathcal{B}_\mathcal{X} \oplus \mathcal{B}_\mathcal{Y}$.

Taking our optimal $(f^*, g^*)$, we can see that

$$\|(f^*, g^*) + (f^0, g^0)\|_{\mathcal{B}_\mathcal{X} \oplus \mathcal{B}_\mathcal{Y}} \geq \|(f^*, g^*)\|_{\mathcal{B}_\mathcal{X} \oplus \mathcal{B}_\mathcal{Y}} \quad \text{for any } (f^0, g^0) \in \mathcal{N}_{\mathcal{X},\mathcal{Y},0} \tag{B.7}$$

thus showing that $(f^*, g^*)$ is orthogonal to the subspace $\mathcal{N}_{\mathcal{X},\mathcal{Y},0}$.

Consider the left and right preimages of $\mathcal{N}_{\mathcal{X},\mathcal{Y},0}$ under $\langle \cdot, \cdot \rangle_{\mathcal{B}_\mathcal{X} \times \mathcal{B}_\mathcal{Y}}$:

$$\langle \cdot, g \rangle_{\mathcal{B}_\mathcal{X} \times \mathcal{B}_\mathcal{Y}}^{-1} [\mathcal{N}_{\mathcal{X},\mathcal{Y},0}] = \Big\{ f \in \mathcal{B}_\mathcal{X} : \quad \left\langle f_{\Phi_\mathcal{Y}(y_j)}, g_{\Phi_\mathcal{X}(x_i)} \right\rangle_{\mathcal{B}_\mathcal{X} \times \mathcal{B}_\mathcal{Y}} = 0;$$

$$i = 1, \dots, n_x; \quad j = 1, \dots, n_y \Big\}; \quad g \in \mathcal{B}_\mathcal{Y}$$

$$\langle f, \cdot \rangle_{\mathcal{B}_\mathcal{X} \times \mathcal{B}_\mathcal{Y}}^{-1} [\mathcal{N}_{\mathcal{X},\mathcal{Y},0}] = \Big\{ f \in \mathcal{B}_\mathcal{X} : \quad \left\langle f_{\Phi_\mathcal{Y}(y_j)}, g_{\Phi_\mathcal{X}(x_i)} \right\rangle_{\mathcal{B}_\mathcal{X} \times \mathcal{B}_\mathcal{Y}} = 0;$$

$$i = 1, \dots, n_x; \quad j = 1, \dots, n_y \Big\}; \quad f \in \mathcal{B}_\mathcal{X}.$$

---

[2]As Fasshauer et al. (2015) note, any Hilbert space is strictly convex and smooth, so it seems reasonable to assume that an RKBS is also strictly convex and smooth.

Since $\langle \cdot, g \rangle_{\mathcal{B}_\mathcal{X} \times \mathcal{B}_\mathcal{Y}}^{-1} [\mathcal{N}_{\mathcal{X},\mathcal{Y},0}] \subseteq \mathcal{B}_\mathcal{X}$ and $\langle f, \cdot \rangle_{\mathcal{B}_\mathcal{X} \times \mathcal{B}_\mathcal{Y}}^{-1} [\mathcal{N}_{\mathcal{X},\mathcal{Y},0}] \subseteq \mathcal{B}_\mathcal{Y}$, we can consider them as normed vector spaces with norms $\| \cdot \|_{\mathcal{B}_\mathcal{X}}$ and $\| \cdot \|_{\mathcal{B}_\mathcal{Y}}$, respectively. From (B.7) and the definition of the direct sum norm (B.3),

$$\|f^* + f^0\|_{\mathcal{B}_\mathcal{X}} \geq \|f^*\|_{\mathcal{B}_\mathcal{X}} \quad \text{for all } f^0 \in \langle \cdot, g \rangle_{\mathcal{B}_\mathcal{X} \times \mathcal{B}_\mathcal{Y}}^{-1} [\mathcal{N}_{\mathcal{X},\mathcal{Y},0}], \text{ for arbitrary } g \in \mathcal{B}_\mathcal{Y} \quad \text{(B.8a)}$$

$$\|g^* + g^0\|_{\mathcal{B}_\mathcal{Y}} \geq \|g^*\|_{\mathcal{B}_\mathcal{Y}} \quad \text{for all } g^0 \in \langle f, \cdot \rangle_{\mathcal{B}_\mathcal{X} \times \mathcal{B}_\mathcal{Y}}^{-1} [\mathcal{N}_{\mathcal{X},\mathcal{Y},0}], \text{ for arbitrary } f \in \mathcal{B}_\mathcal{X}. \quad \text{(B.8b)}$$

We can then use (B.8) and Lemma 1 to say

$$\langle f, \iota(f^*) \rangle_{\mathcal{B}_\mathcal{X}} = 0 \quad \text{for all } f \in \langle \cdot, g \rangle_{\mathcal{B}_\mathcal{X} \times \mathcal{B}_\mathcal{Y}}^{-1} [\mathcal{N}_{\mathcal{X},\mathcal{Y},0}], \text{ for arbitrary } g \in \mathcal{B}_\mathcal{Y}$$

$$\langle g, \iota(g^*) \rangle_{\mathcal{B}_\mathcal{Y}} = 0 \quad \text{for all } g \in \langle f, \cdot \rangle_{\mathcal{B}_\mathcal{X} \times \mathcal{B}_\mathcal{Y}}^{-1} [\mathcal{N}_{\mathcal{X},\mathcal{Y},0}], \text{ for arbitrary } f \in \mathcal{B}_\mathcal{X}$$

which means

$$\iota(f^*) \in \left( \langle \cdot, g \rangle_{\mathcal{B}_\mathcal{X} \times \mathcal{B}_\mathcal{Y}}^{-1} [\mathcal{N}_{\mathcal{X},\mathcal{Y},0}] \right)^\perp \quad \text{for all } g \in \mathcal{B}_\mathcal{Y} \quad \text{(B.9a)}$$

$$\iota(g^*) \in \left( \langle f, \cdot \rangle_{\mathcal{B}_\mathcal{X} \times \mathcal{B}_\mathcal{Y}}^{-1} [\mathcal{N}_{\mathcal{X},\mathcal{Y},0}] \right)^\perp \quad \text{for all } f \in \mathcal{B}_\mathcal{X}. \quad \text{(B.9b)}$$

From (B.9a) and (3a)-(3b),

$$\begin{aligned}
f^* \in \bigcup_{g \in \mathcal{B}_\mathcal{Y}} \langle \cdot, g \rangle_{\mathcal{B}_\mathcal{X} \times \mathcal{B}_\mathcal{Y}}^{-1} [\mathcal{N}_{\mathcal{X},\mathcal{Y},0}] &= \left\{ f \in \mathcal{B}_\mathcal{X} : \left\langle f_{\Phi_\mathcal{Y}(y_j)}, g_{\Phi_\mathcal{X}(x_i)} \right\rangle_{\mathcal{B}_\mathcal{X} \times \mathcal{B}_\mathcal{Y}} = 0; \quad g \in \mathcal{B}_\mathcal{Y}; \right. \\
&\qquad\qquad i = 1, \ldots, n_x; \quad j = 1, \ldots, n_y \Big\} \\
&= \left\{ f \in \mathcal{B}_\mathcal{X} : \left\langle f_{\Phi_\mathcal{Y}(y_j)}, h \right\rangle_{\mathcal{B}_\mathcal{X} \times \mathcal{B}_\mathcal{Y}} = 0 \right. \\
&\qquad\qquad h \in \operatorname{span} \{ \kappa(x_i, \cdot); \quad i = 1, \ldots, n_x \} ; \\
&\qquad\qquad j = 1, \ldots, n_y \Big\} \\
&= {}^\perp \operatorname{span} \{ \kappa(x_i, \cdot); \quad i = 1, \ldots, n_x \} . \quad \text{(B.10)}
\end{aligned}$$

And from (B.9b) and (3c)-(3d),

$$\begin{aligned}
g^* \in \bigcup_{f \in \mathcal{B}_\mathcal{X}} \langle f, \cdot \rangle_{\mathcal{B}_\mathcal{X} \times \mathcal{B}_\mathcal{Y}}^{-1} [\mathcal{N}_{\mathcal{X},\mathcal{Y},0}] &= \left\{ g \in \mathcal{B}_\mathcal{Y} : \left\langle f_{\Phi_\mathcal{Y}(y_j)}, g_{\Phi_\mathcal{X}(x_i)} \right\rangle_{\mathcal{B}_\mathcal{X} \times \mathcal{B}_\mathcal{Y}} = 0; \quad f \in \mathcal{B}_\mathcal{X}; \right. \\
&\qquad\qquad i = 1, \ldots, n_x; \quad j = 1, \ldots, n_y \Big\} \\
&= \left\{ g \in \mathcal{B}_\mathcal{Y} : \left\langle h', f_{\Phi_\mathcal{Y}(y_j)} \right\rangle_{\mathcal{B}_\mathcal{X} \times \mathcal{B}_\mathcal{Y}} = 0 \right. \\
&\qquad\qquad h' \in \operatorname{span} \{ \kappa(\cdot, y_j); \quad j = 1, \ldots, n_y \} ; \\
&\qquad\qquad i = 1, \ldots, n_x \Big\} \\
&= {}^\perp \operatorname{span} \{ \kappa(\cdot, y_j); \quad y = 1, \ldots, n_y \} . \quad \text{(B.11)}
\end{aligned}$$

Combining (B.9a) and (B.10), we get

$$\iota(f^*) \in \left( {}^\perp \operatorname{span}\{ \kappa(x_i, \cdot); \quad i = 1, \ldots, n_x \} \right)^\perp = \overline{\operatorname{span}}\{ \kappa(x_i, \cdot); \quad i = 1, \ldots, n_x \} \quad \text{(B.12)}$$

and by combining (B.9b) and (B.11), we get

$$\iota(g^*) \in \left( {}^\perp \operatorname{span}\{ \kappa(\cdot, y_j); \quad j = 1, \ldots, n_y \} \right)^\perp = \overline{\operatorname{span}}\{ \kappa(\cdot, y_j); \quad j = 1, \ldots, n_y \} \quad \text{(B.13)}$$

where in both (B.12) and (B.13) we use Proposition 2.6.6 of Megginson (1998) regarding properties of annihilators.

From (B.12) and (B.13) we can write that there exist two sets of parameters $\xi_1, \ldots, \xi_{n_x} \in \mathbb{R}$ and $\zeta_1, \ldots, \zeta_{n_y} \in \mathbb{R}$ such that

$$\iota(f^*) = \sum_{i=1}^{n_x} \xi_i \kappa(x_i, \cdot) \qquad\qquad \iota(g^*) = \sum_{j=1}^{n_y} \zeta_j \kappa(\cdot, y_j)$$

thus proving the claim. $\qquad\qquad\square$

### B.3 MAIN PROOF

Before beginning the proof, we state the following lemma regarding the existence of solutions of convex optimization problems on Banach spaces:

**Lemma 4** (Ekeland & Témam, 1999, Chapter II, Proposition 1.2). *Let $\mathcal{B}$ be a reflexive Banach space and $\mathcal{S}$ a closed, convex, and bounded (with respect to $\|\cdot\|_{\mathcal{B}}$) subset of $\mathcal{B}$. Let $f : \mathcal{S} \to \mathbb{R} \cup \{+\infty\}$ be a convex function with a closed epigraph (i.e., it satisfies the condition that $\forall c \in \mathbb{R} \cup \{+\infty\}$, the set $\{v \in \mathcal{S} : f(v) \leq c\}$ is closed). Then, the optimization problem*

$$\inf_{v \in \mathcal{S}} f(v)$$

*has at least one solution.*

Xu & Ye (2019) and Lin et al. (2019) also reference Ekeland & Turnbull (1983) as a source for Lemma 4.

We now restate Theorem 1 with the conditions on $\mathcal{B}_{\mathcal{X}}$ and $\mathcal{B}_{\mathcal{Y}}$ filled in.

**Theorem 1, Revisited.** *Suppose we have a kernel learning problem of the form in* (11). *Let $\kappa : \mathcal{X} \times \mathcal{Y} \to \mathbb{R}$, $\kappa(x_i, y_i) = \langle \Phi_{\mathcal{X}}(x_i), \Phi_{\mathcal{Y}}(y_i) \rangle_{\mathcal{F}_{\mathcal{X}} \times \mathcal{F}_{\mathcal{Y}}} = \langle f_{\Phi_{\mathcal{Y}}(y)}, g_{\Phi_{\mathcal{X}}(x)} \rangle_{\mathcal{B}_{\mathcal{X}} \times \mathcal{B}_{\mathcal{Y}}}$ be a reproducing kernel satisfying Definitions 1 and 2. Assume that $\{\kappa(x_i, \cdot)\}_{i=1}^{n_x}$ is linearly independent in $\mathcal{B}_{\mathcal{Y}}$ and that $\{\kappa(\cdot, y_j)\}_{j=1}^{n_y}$ is linearly independent in $\mathcal{B}_{\mathcal{X}}$. Assume also that $\mathcal{B}_{\mathcal{X}}$ and $\mathcal{B}_{\mathcal{Y}}$ are reflexive, strictly convex, and smooth. Then, the regularized empirical risk minimization problem* (11) *has a unique solution pair $(f^*, g^*)$, with the property that*

$$\iota(f^*) = \sum_{i=1}^{n_x} \xi_i \kappa(x_i, \cdot) \qquad \iota(g^*) = \sum_{j=1}^{n_y} \zeta_j \kappa(\cdot, y_j).$$

*where $\xi_i, \zeta_j \in \mathbb{R}$.*

*Proof.* As before, we begin by proving existence and uniqueness of the solution pair $(f^*, g^*)$.

We first prove uniqueness using some basic facts about convexity. Assume that there exist two distinct minimizers $(f_1^*, g_1^*), (f_2^*, g_2^*) \in \mathcal{B}_{\mathcal{X}} \oplus \mathcal{B}_{\mathcal{Y}}$. Define $(f_3^*, g_3^*) = \frac{1}{2}[(f_1^*, g_1^*) + (f_2^*, g_2^*)]$. Then, since $\mathcal{B}_{\mathcal{X}}$ and $\mathcal{B}_{\mathcal{Y}}$ are strictly convex, we have

$$\|f_3^*\|_{\mathcal{B}_{\mathcal{X}}} = \left\|\frac{1}{2}(f_1^* + f_2^*)\right\|_{\mathcal{B}_{\mathcal{X}}} < \frac{1}{2}\|f_1^*\|_{\mathcal{B}_{\mathcal{X}}} + \frac{1}{2}\|f_2^*\|_{\mathcal{B}_{\mathcal{X}}}$$

$$\|g_3^*\|_{\mathcal{B}_{\mathcal{Y}}} = \left\|\frac{1}{2}(g_1^* + g_2^*)\right\|_{\mathcal{B}_{\mathcal{Y}}} < \frac{1}{2}\|g_1^*\|_{\mathcal{B}_{\mathcal{Y}}} + \frac{1}{2}\|g_2^*\|_{\mathcal{B}_{\mathcal{Y}}}$$

and since $R_{\mathcal{X}}$ and $R_{\mathcal{Y}}$ are convex and strictly increasing,

$$R_{\mathcal{X}}(\|f_3^*\|_{\mathcal{B}_{\mathcal{X}}}) = R_{\mathcal{X}}\left(\left\|\frac{1}{2}(f_1^* + f_2^*)\right\|_{\mathcal{B}_{\mathcal{X}}}\right) < R_{\mathcal{X}}\left(\frac{1}{2}\|f_1^*\|_{\mathcal{B}_{\mathcal{X}}} + \frac{1}{2}\|f_2^*\|_{\mathcal{B}_{\mathcal{X}}}\right)$$

$$\leq \frac{1}{2}R_{\mathcal{X}}(\|f_1^*\|_{\mathcal{B}_{\mathcal{X}}} + \frac{1}{2}R_{\mathcal{X}}(\|f_2^*\|_{\mathcal{B}_{\mathcal{X}}})$$

and

$$R_{\mathcal{Y}}(\|g_3^*\|_{\mathcal{B}_{\mathcal{Y}}}) = R_{\mathcal{Y}}\left(\left\|\frac{1}{2}(g_1^* + g_2^*)\right\|_{\mathcal{B}_{\mathcal{Y}}}\right) < R_{\mathcal{Y}}\left(\frac{1}{2}\|g_1^*\|_{\mathcal{B}_{\mathcal{Y}}} + \frac{1}{2}\|g_2^*\|_{\mathcal{B}_{\mathcal{Y}}}\right)$$

$$\leq \frac{1}{2}R_{\mathcal{Y}}(\|g_1^*\|_{\mathcal{B}_{\mathcal{Y}}} + \frac{1}{2}R_{\mathcal{Y}}(\|g_2^*\|_{\mathcal{B}_{\mathcal{Y}}}).$$

Consider the regularized empirical risk minimization cost function (11)

$$\mathcal{T}(f, g) = \mathcal{L}(f, g) + \lambda_{\mathcal{X}} R_{\mathcal{X}}(\|f\|_{\mathcal{B}_{\mathcal{X}}}) + \lambda_{\mathcal{Y}} R_{\mathcal{Y}}(\|g\|_{\mathcal{B}_{\mathcal{Y}}})$$

where we use the shorthand

$$\mathcal{L}(f, g) = \frac{1}{n_x n_y} \sum_{i,j} L\left(x_i, y_j, z_{ij}, \langle f_{\Phi_{\mathcal{Y}}(y)}, g_{\Phi_{\mathcal{X}}(x)} \rangle_{\mathcal{B}_{\mathcal{X}} \times \mathcal{B}_{\mathcal{Y}}}\right).$$

We have that $R_{\mathcal{X}}(\|\cdot\|_{\mathcal{B}_{\mathcal{X}}})$ and $R_{\mathcal{Y}}(\|\cdot\|_{\mathcal{B}_{\mathcal{Y}}})$ are both convex via identities about composition of convex functions. The function $\mathcal{L}(f,g)$ is also convex since all the functions in the summand are convex in $f$ and $g$.

Then, since we have assumed that $\mathcal{T}(f_1^*, g_1^*) = \mathcal{T}(f_2^*, g_2^*)$, by plugging in some of the above inequalities we can write

$$
\begin{aligned}
\mathcal{T}(f_3^*, g_3^*) &= \mathcal{T}\left(\frac{1}{2}\left[(f_1^*, g_1^*) + (f_2^*, g_2^*)\right]\right) \\
&= \mathcal{L}\left(\frac{1}{2}\left[(f_1^*, g_1^*) + (f_2^*, g_2^*)\right]\right) + R_{\mathcal{X}}\left(\left\|\frac{1}{2}(f_1^* + f_2^*)\right\|_{\mathcal{B}_{\mathcal{X}}}\right) \\
&\qquad\qquad + R_{\mathcal{Y}}\left(\left\|\frac{1}{2}(g_1^* + g_2^*)\right\|_{\mathcal{B}_{\mathcal{Y}}}\right) \\
&< \frac{1}{2}\mathcal{L}(f_1^*, g_1^*) + \frac{1}{2}\mathcal{L}(f_2^*, g_2^*) + \frac{1}{2}R_{\mathcal{X}}(\|f_1^*\|_{\mathcal{B}_{\mathcal{X}}}) + \frac{1}{2}R_{\mathcal{X}}(\|f_2^*\|_{\mathcal{B}_{\mathcal{X}}}) \\
&\qquad\qquad + \frac{1}{2}R_{\mathcal{Y}}(\|f_1^*\|_{\mathcal{B}_{\mathcal{Y}}}) + \frac{1}{2}R_{\mathcal{Y}}(\|f_2^*\|_{\mathcal{B}_{\mathcal{Y}}}) \\
&= \frac{1}{2}\mathcal{T}(f_1^*, g_1^*) + \frac{1}{2}\mathcal{T}(f_2^*, g_2^*) \\
&= \mathcal{T}(f_1^*, g_1^*)
\end{aligned}
$$

contradicting that $(f_1^*, g_1^*)$ is a minimizer, and thus showing uniqueness of the solution.

We now prove existence via Lemma 4. We already know that $\mathcal{T}(\cdot)$ is convex. From the bilinearity of $\langle\cdot,\cdot\rangle_{\mathcal{B}_{\mathcal{X}}\times\mathcal{B}_{\mathcal{Y}}}$ and the convexity of $L$, $L$ is continuous in $f$ and $g$. Since the regularization functions $R_{\mathcal{X}}$ and $R_{\mathcal{Y}}$ are convex and strictly increasing, is follows that the functions $R_{\mathcal{X}}(\|f\|_{\mathcal{B}_{\mathcal{X}}})$ and $R_{\mathcal{Y}}(\|g\|_{\mathcal{B}_{\mathcal{Y}}})$ are continuous in $f$ and $g$, respectively. Thus, $\mathcal{T}(f,g)$ is continuous. Consider the set

$$
\mathcal{E} = \{(f,g) \in \mathcal{B}_{\mathcal{X}} \oplus \mathcal{B}_{\mathcal{Y}} : \mathcal{T}(f,g) \leq \mathcal{T}(0,0)\}.
$$

The set $\mathcal{E}$ is nonempty (it contains at least $(0,0)$), and we can see that

$$
\begin{aligned}
\|f,g\|_{\mathcal{B}_{\mathcal{X}}\oplus\mathcal{B}_{\mathcal{Y}}} &= \|f\|_{\mathcal{B}_{\mathcal{X}}} + \|g\|_{\mathcal{B}_{\mathcal{Y}}} \\
&\leq R_{\mathcal{X}}^{-1}(\mathcal{T}(f,0)) + R_{\mathcal{Y}}^{-1}(\mathcal{T}(0,g))
\end{aligned}
$$

showing that $\mathcal{E}$ is bounded. So, by Lemma 4, we are guaranteed the existence of an optimal solution $(f^*, g^*)$.

Pick any $(f,g) \in \mathcal{B}_{\mathcal{X}} \times \mathcal{B}_{\mathcal{Y}}$ and consider the set

$$
D_{f,g} = \left\{\left(x_i, y_j, \left\langle f_{\Phi_{\mathcal{Y}}(y_j)}, g_{\Phi_{\mathcal{X}}(x_i)}\right\rangle_{\mathcal{B}_{\mathcal{X}}\times\mathcal{B}_{\mathcal{Y}}}\right) : \quad i = 1,\ldots,n_x; \quad j = 1,\ldots,n_y\right\}
$$

i.e., the set of pairs of points $(x_i, y_j)$ along with the value that the function pair $(f,g)$ maps to via the bilinear form at the pair of points $(x_i, y_j)$.

From Lemma 3, there exists an element $(f', g') \in \mathcal{B}_{\mathcal{X}} \times \mathcal{B}_{\mathcal{Y}}$ such that $(f', g')$ interpolates $D_{f,g}$ perfectly, i.e.,

$$
\left\langle f_{\Phi_{\mathcal{Y}}(y_j)}, g_{\Phi_{\mathcal{X}}(x_i)}\right\rangle_{\mathcal{B}_{\mathcal{X}}\times\mathcal{B}_{\mathcal{Y}}} = \left\langle f'_{\Phi_{\mathcal{Y}}(y_j)}, g'_{\Phi_{\mathcal{X}}(x_i)}\right\rangle_{\mathcal{B}_{\mathcal{X}}\times\mathcal{B}_{\mathcal{Y}}}; \quad i = 1,\ldots,n_x; \quad j = 1,\ldots,n_y
$$

whose Gâteaux derivatives of norms satisfy

$$
\begin{aligned}
\iota(f') &\in \overline{\mathrm{span}}\{\kappa(x_i, \cdot); \quad i = 1,\ldots,n_x\} \\
\iota(g') &\in \overline{\mathrm{span}}\{\kappa(\cdot, y_j); \quad j = 1,\ldots,n_y\}.
\end{aligned}
$$

Further, this element $(f', g')$ obtains the minimum-norm interpolation of $D_{f,g}$, i.e.,

$$
\|f', g'\|_{\mathcal{B}_{\mathcal{X}}\oplus\mathcal{B}_{\mathcal{Y}}} \leq \|f,g\|_{\mathcal{B}_{\mathcal{X}}\oplus\mathcal{B}_{\mathcal{Y}}}.
$$

This last fact implies

$$
T(f', g') \leq T(f,g).
$$

Therefore, the unique optimal solution $(f^*, g^*)$ also satisfies

$$\iota(f^*) \in \overline{\text{span}}\{\kappa(x_i, \cdot); \quad i = 1, \ldots, n_x\}$$
$$\iota(g^*) \in \overline{\text{span}}\{\kappa(\cdot, y_j); \quad j = 1, \ldots, n_y\}$$

which implies that suitable parameters $\xi_1, \ldots, \xi_{n_x} \in \mathbb{R}$ and $\zeta_1, \ldots, \zeta_{n_y} \in \mathbb{R}$ exist such that

$$\iota(f^*) = \sum_{i=1}^{n_x} \xi_i \kappa(x_i, \cdot) \qquad \iota(g^*) = \sum_{j=1}^{n_y} \zeta_j \kappa(\cdot, y_j)$$

proving the claim. $\qquad\square$

## C  PROOF OF THEOREM 2

First, we state the following well-known lemma.

**Lemma 5.** *For any two compact Hausdorff spaces $\mathcal{X}$ and $\mathcal{Y}$, continuous function $\kappa : \mathcal{X} \times \mathcal{Y} \to \mathbb{R}$, and $\epsilon > 0$, there exists an integer $d > 0$ and continuous functions $\phi_\ell : \mathcal{X} \to \mathbb{R}$, $\psi_\ell : \mathcal{Y} \to \mathbb{R}$, $\ell = 1, \ldots, d$ such that*

$$\left| \kappa(x, y) - \sum_{\ell=1}^{d} \phi_\ell(x)\psi_\ell(y) \right| < \epsilon \quad \forall \, x \in \mathcal{X}, y \in \mathcal{Y}.$$

*Proof.* The product space of two compact spaces $\mathcal{X}$ and $\mathcal{Y}$, $\mathcal{X} \times \mathcal{Y}$, is of course compact by Tychonoff's theorem. Consider the algebra

$$\mathbb{A} = \left\{ \hat{f} : \hat{f}(x, y) = \sum_{\ell=1}^{d} \phi_\ell(x)\psi_\ell(y), \ x \in \mathcal{X}, y \in \mathcal{Y} \right\}.$$

It is easy to show (i) that $\mathbb{A}$ is an algebra, (ii) that $\mathbb{A}$ is a subalgebra of the real-valued continuous functions on $\mathcal{X} \times \mathcal{Y}$, and (iii) that $\mathbb{A}$ separates points. Then, combining the aforementioned facts, by the Stone-Weierstrass theorem $\mathbb{A}$ is dense in the set of real-valued continuous functions on $\mathcal{X} \times \mathcal{Y}$. $\quad\square$

*Remark* 9. In addition to helping us prove Theorem 2 below, Lemma 5 also serves as somewhat of an analog to Mercer's theorem for the more general case of asymmetric, non-PSD kernels. It is however weaker than Mercer's theorem in that the non-PSD nature of $\kappa$ means that the functions in the sum cannot be considered as eigenfunctions (with $\phi_\ell = \psi_\ell$) with associated nonnegative eigenvalues.

Now we proceed to the proof of Theorem 2.

*Proof.* To keep the equations from becoming too cluttered, below we use $\boldsymbol{q}(\boldsymbol{t}), \boldsymbol{k}(\boldsymbol{s}), \boldsymbol{\phi}(\boldsymbol{t}), \boldsymbol{\psi}(\boldsymbol{s}) \in \mathbb{R}^d$ as the vector concatenation of the scalar functions $\{q_\ell(\boldsymbol{t})\}, \{k_\ell(\boldsymbol{s})\}, \{\phi_\ell(\boldsymbol{t})\}$, and $\{\psi_\ell(\boldsymbol{s})\}$, $\ell = 1, \ldots, d$, respectively. All sup norms are with respect to $\mathcal{X} \times \mathcal{Y}$.

Our proof proceeds similarly to the proof of Theorem 5.1 of Okuno et al. (2018). We generalize their theorem and proof to non-Mercer kernels and simplify some intermediate steps. First, by applying Lemma 5, we can write that for any $\epsilon_1$, there is a $d$ such that

$$\left\| \kappa - \boldsymbol{\phi}^\mathsf{T} \boldsymbol{\psi} \right\|_{\text{sup}} < \epsilon_1 \tag{C.1}$$

Now we consider the approximation of $\phi_\ell$ and $\psi_\ell$ by $q_\ell$ and $k_\ell$, respectively. By the universal approximation theorem of multilayer neural networks (Cybenko, 1989; Hornik et al., 1989; Funahashi, 1989; Attali & Pagès, 1997, etc.), we know that for any functions $\phi_\ell : \mathcal{X} \to \mathbb{R}$, $\psi_\ell : \mathcal{Y} \to \mathbb{R}$ and scalar $\epsilon_2 > 0$, there is an integer $m > 0$ such that if $q_\ell : \mathcal{X} \to \mathbb{R}$ and $k_\ell : \mathcal{Y} \to \mathbb{R}$ are two-layer neural networks with $m$ hidden units, then

$$\left\| \phi_\ell - q_\ell \right\|_{\text{sup}} < \epsilon_2 \quad \text{and} \quad \left\| \psi_\ell - k_\ell \right\|_{\text{sup}} < \epsilon_2 \tag{C.2}$$

for all $\ell$. Now, beginning from (14), we can write

$$\left\| \kappa - \boldsymbol{q}^\mathsf{T} \boldsymbol{k} \right\|_{\text{sup}} \leq \left\| \kappa - \boldsymbol{\phi}^\mathsf{T} \boldsymbol{\psi} \right\|_{\text{sup}} + \left\| \boldsymbol{\phi}^\mathsf{T} \boldsymbol{\psi} - \boldsymbol{k}^\mathsf{T} \boldsymbol{q} \right\|_{\text{sup}} \tag{C.3}$$

by the triangle inequality. Examining the second term of the RHS of (C.3),

$$\left\|\boldsymbol{\phi}^\mathsf{T}\boldsymbol{\psi} - \boldsymbol{q}^\mathsf{T}\boldsymbol{k}\right\|_{\sup} = \left\|\boldsymbol{\phi}^\mathsf{T}(\boldsymbol{\psi} - \boldsymbol{k}) + (\boldsymbol{\phi} - \boldsymbol{q})^\mathsf{T}\boldsymbol{k}\right\|_{\sup}$$
$$\leq \left\|\boldsymbol{\phi}^\mathsf{T}(\boldsymbol{\psi} - \boldsymbol{k})\right\|_{\sup} + \left\|(\boldsymbol{\phi} - \boldsymbol{q})^\mathsf{T}\boldsymbol{k}\right\|_{\sup}$$
$$\leq \left\|\boldsymbol{\phi}\right\|_{\sup}\left\|\boldsymbol{\psi} - \boldsymbol{k}\right\|_{\sup} + \left\|\boldsymbol{\phi} - \boldsymbol{q}\right\|_{\sup}\left\|\boldsymbol{k}\right\|_{\sup} \qquad (\text{C.4})$$

where the first inequality uses the triangle inequality and the second uses the Cauchy-Schwarz inequality. Finally, we can combine (C.1)-(C.4) to write

$$\left\|\kappa - \boldsymbol{q}^\mathsf{T}\boldsymbol{k}\right\|_{\sup} \leq \left\|\kappa - \boldsymbol{\phi}^\mathsf{T}\boldsymbol{\psi}\right\|_{\sup} + \left\|\boldsymbol{\phi}\right\|_{\sup}\left\|\boldsymbol{\psi} - \boldsymbol{k}\right\|_{\sup} + \left\|\boldsymbol{\phi} - \boldsymbol{q}\right\|_{\sup}\left\|\boldsymbol{k}\right\|_{\sup}$$
$$< \epsilon_1 + d\epsilon_2(\left\|\boldsymbol{\phi}\right\|_{\sup} + \left\|\boldsymbol{\psi}\right\|_{\sup} + d\epsilon_2).$$

Picking $\epsilon_1$ and $\epsilon_2$ appropriately, e.g. $\epsilon_1 = \epsilon/2$ and $\epsilon_2 \leq \frac{\sqrt{d^2((\|\boldsymbol{\phi}\|_{\sup} + \|\boldsymbol{\psi}\|_{\sup})^2 + 2\epsilon)}}{2d^2}$, completes the proof. $\qquad\square$

## D  EXPERIMENT IMPLEMENTATION DETAILS

**Datasets**  The 2014 International Workshop on Spoken Language Translation (IWSLT14) machine translation dataset is a dataset of transcriptions of TED talks (and translations of those transcriptions). We use the popular German to English subset of the dataset. We use the 2014 "dev" set as our test set, and a train/validation split suggested in demo code for `fairseq` (Ott et al., 2019), where every 23rd line in the IWSLT14 training data is held out as a validation set.

The 2014 ACL Workshop on Statistical Machine Translation (WMT14) dataset is a collection of European Union Parliamentary proceedings, news stories, and web text with multiple translations. We use `newstest2013` as our validation set and `newstest2014` as our test set.

The Stanford Sentiment Treebank (SST) (Socher et al., 2013) is a sentiment analysis dataset with sentences taken from movie reviews. We use two standard subtasks: binary classification (SST-2) and fine-grained classification (SST-5). SST-2 is a subset of SST-5 with neutral-labeled sentences removed. We use the standard training/validation/testing splits, which gives splits of 6920/872/1821 on SST-2 and 8544/1101/2210 on SST-5.

**Data Preprocessing**  On both translation datasets, we use `sentencepiece`[3] to tokenize and train a byte-pair encoding (Sennrich et al., 2016) on the training set. We use a shared BPE vocabulary across the target and source languages. Our resulting BPE vocabulary size is 8000 for IWSLT14 DE-EN and 32000 for WMT14 EN-FR.

For SST, we train a `sentencepiece` BPE for each subtask separately, obtaining BPE vocabularies of size 7465 for SST-2 and 7609 for SST-5.

**Models**  Our models are written in Pytorch (Paszke et al., 2019). We make use of the `Fairseq` (Ott et al., 2019) library for training and evaluation.

In machine translation, we use 6 Transformer layers in both the encoder and decoder. Both Transformer sublayers (attention and the two fully-connected layers) have a residual connection with the "pre-norm" (Wang et al., 2019b) ordering of Layer normalization -> Attention or FC -> ReLU -> Add residual. We use an embedding dimension of 512 for the learned token embeddings. For IWSLT14, the attention sublayers use 4 heads with a per-head dimension $d$ of 128 and the fully-connected sublayers have a hidden dimension of 1024. For WMT14, following Vaswani et al. (2017)'s "base" model, the attention layers have 8 heads with a per-head dimension $d$ of 64 and the fully-connected sublayers have a hidden dimension of 2048.

For SST, we use a very small, encoder-only, Transformer variant, with only two Transformer layers. The token embedding dimension is 64, each Transformer self-attention sublayer has 4 heads with per-head dimension $d$ of 16, and the fully-connected sublayers have a hidden dimension of 128. To produce a sentence classification, the output of the second Transformer layer is average-pooled over

---

[3]https://github.com/google/sentencepiece

the non-padding tokens, then passed to a classification head. This classification head is a two-layer neural network with hidden dimension 64 and output dimension equal to the number of classes; this output vector becomes the class logits.

**Training**    We train with the Adam optimizer (Kingma & Ba, 2015). Following Vaswani et al. (2017), for machine translation we set the Adam parameters $\beta_1 = 0.9, \beta_2 = 0.98$.

On IWSLT14 DE-EN, we schedule the learning rate to begin at 0.001 and then multiply by a factor of 0.1 when the validation BLEU does not increase for 3 epochs. FOR WMT14 EN-FR, we decay proportionally to the inverse square root of the update step using `Fairseq`'s implementation. For both datasets, we also use a linear warmup on the learning rate from 1e-7 to 0.001 over the first 4000 update steps.

On IWSLT14 DE-EN, we end training when the BLEU score does not improve for 7 epochs on the validation set. On WMT14 EN-FR, we end training after 100k gradient updates (inclusive of the warmup stage), which gives us a final learning rate of 0.0002. We train on the cross-entropy loss and employ label smoothing of 0.1. We use minibatches with a maximum of about 10k source tokens on IWSLT14 DE-EN and 25k on WMT14 EN-FR Also on WMT14, we ignore sentences with more than 1024 tokens.

For both SST subtasks, we also use a linear warmup from 1e-7 over 4000 warmup steps, but use an initial post-warmup learning rate of 0.0001. Similar to IWSLT14, we decay the learning rate by multiplying by 0.1 when the validation accuracy does not increase for 3 epochs, and end training when the validation accuracy does not improve for 8 epochs.

**Evaluation for machine translation**    Following Vaswani et al. (2017), we use beam-search decoding, with a beam length of 4 and a length penalty of 0.6, to generate sentences for evaluation. We use `sacrebleu` (Post, 2018) to generate BLEU scores. We report whole-word case-sensitive BLEU.

