# OpenReview forum: "Transformers are Deep Infinite-Dimensional Non-Mercer Binary Kernel Machines"
_ICLR.cc/2021/Conference — Reject_

### Official Review · AnonReviewer4 · 2020-10-16
**Transformer models are non-mercer binary kernel machines on reproducing kernel Banach spaces**

**Rating:** 6
**Confidence:** 4

**Review:**

Review: This paper demonstrates that transformer models with dot-product attention-based scores are inherently
learning feature representations in reproducing kernel Banach spaces. Under some mild regularity conditions,
the authors demonstrate that the regularized empirical minimization solutions are unique, (i.e. linear independence
assumptions in the two reproducing kernel Banach spaces corresponding to the targets and the sources) and that
using two-layer neural network with appropriate number of hidden layer units can achieve universal approximation
property.

Overall reasons for score:
I am leaning toward arguing for acceptance. The paper establishes a connection between a fairly recent method a classical method. It would have been nice to see more experimental results (see below).

+Positives:

+ The paper for the most part is clearly written and establishes a connection between the recent method and the classical
method. I have skimmed through most of the proofs and they seem to be correct.


Concerns:

- The experimental section is fairly limited since the majority of the paper focuses on more theoretical aspects of transformers.
In particular, it would be interesting to see the authors expand upon whether exponentiated dot products are better on
more datasets.

- The theoretical contributions in this paper are interesting but mainly pieces together results.

Minor comments:

* Proposition 1 has typos in Equation 7a and 7b. (q_l)^n -> (q_l)^{p_l} and (k_l)^n -> (k_l)^{p_l}

---

> ### Author Response · Authors · 2020-11-16
> **Response to reviewer 4**
>
> Thank you for the review.
>
> Regarding the request for additional experimental results, we have updated the PDF with additional results on the considerably larger WMT14 English-French dataset. As we mention in the expanded section, the results agree with the IWSLT14 German-English experiment, in that the infinite dimensionality of the kernel seems key to performance.
> We are working on adding some experiments on text classification datasets to the paper.
>
> Thank you for pointing out the typo.

---

### Official Review · AnonReviewer2 · 2020-10-27
**a two-fold contribution - transformers and kernel machines.**

**Rating:** 7
**Confidence:** 4

**Review:**

The paper aims at making a link between kernels in RKBS (indefinite and asymmetric kernels) and the dot-product attention of Transformers. The paper contains several contributions on top of this link : it provides a novel kernel machine that can deal with data from 2 distinct input domains and a cross domain output, it show that Transformer can learn such kernels, and it also give hints on what make Transformer efficient.

The paper manages to present different backgrounds (transformers and exotic kernels) to mix them in a quite clear manner. Notations from two worlds are respected such that, as far as I can say, people from each side can catch things quickly.
From the transformer's perspective, being able to plug any well designed kernel in place of the dot-product attention can have some interesting practical application. The observation that the efficiency of transformer could be linked to the infinite feature map brought be some kernel shapes is appealing but quite weak.
From the kernel machine's perspective, the proposed kernel machine in RKBS is elegant and comes with solid theoretical proofs. It might have been a paper by itself.
The choice of content in paper/in supplementary seems good to me.

I vote for accepting the paper, as I find the idea interesting, I can see some applications and the theoretical part seems correct to me. My main concern is about section 6, which contains only one experiment. It illustrates the fact that one can change the kernel in transformers, but not much more.
I also have difficulties to see how section 5.2 is done in practice, as I find the description of implementation details do not consider those aspects : I would be happy to have more details on the algorithms

- Details :
* section 6 : I suggest the order of kernels be the same in table and in text, it would be easier to follow.
* section 5.2 : I think this reference (https://doi.org/10.1016/j.patcog.2017.06.003) introduces Nystrom for RKKS a couple of years earlier.

---

> ### Author Response · Authors · 2020-11-16
> **Response to reviewer 2**
>
> Thank you for the positive review.
>
> We have updated the paper with an additional experiment on the much larger WMT14 English-French translation dataset. As we note in our expanded experiment discussion, the results seem to align with those on the smaller IWSLT14 German-English dataset. We are currently working towards adding experiments on text classification datasets as well.
>
> Regarding section 5.2: our intention in that section was to briefly mention the Nystrom method as background information, then state that the Transformer itself is another way to approximate a kernel method. In particular, the natural log of the exponentiated query-key kernel
> We see that we made some consistency mistakes between how we described the kernel in, e.g., equation (9) and how we use $\kappa$ in Theorem 2. We have updated the paper to try and clarify this, as well as added some more explanation to this section. Please let if know if you still have concerns.
>
> Thank you for the advice on reformatting our results table and letting us know about this additional reference regarding the Nystrom method in RKKS's. We have added this reference.

---

### Official Review · AnonReviewer1 · 2020-10-27
**ICLR 2021 Conference Paper2896 AnonReviewer1**

**Rating:** 4
**Confidence:** 4

**Review:**

The paper aims at providing a mathematical structure for explaining the mechanism behind the attention block characteristic to transformers. The focus of the paper is on the scaled dot-product attention, reviewed in Eq. (1). In my understanding, the whole mechanism can be seen as an instance of the set kernel. The inputs are bags of items, where an item is denoted with $s_j$. The items are embedded into some feature space via matrix multiplication $W^V s_j$. The set kernel representation of a bag is obtained by weighted averaging of item embeddings, where the item-specific weight is the output of an exponential family model. The latter model is obtained by combining embeddings of items and corresponding context vectors, denoted with $t_i$ (see Eq. 1 for more details).

The paper in itself does not explain this mechanism as a whole but focuses on the un-normalized exponential family model that provides importance weights in the set kernel embedding. In particular, the main focus of the work is to find a bilinear form and, thus, a mathematical structure that could give rise to non-symmetric similarity function defining the un-normalized importance weights in the set kernel described above. More formally, the work seeks a kernel function
$$
k(t,s) = \exp(\frac{(W^Qt)^{\top}(W^Ks)}{\sqrt{d}}) \ ,
$$
where $d$ is the rank of W-matrices.

This is achieved by introducing the so called reproducing kernel Banach space (Sections 3 and 4). Following this, the paper introduces a form of regularized risk minimization problem in reproducing kernel Banach spaces. I fail to see a direct link to transformers and backpropagation used in training of such models. The conclusion is that transformers learn a kernel or similarity function that can be assigned to a reproducing kernel Banach space. I disagree with this because the whole attention mechanism is a set kernel, with the supplied bilinear form amounting to importance weights only. The section 5 concludes with a representer theorem and regularized risk minimization problem in reproducing kernel Banach spaces. Again, this is a completely disconnected part of the paper from the introduction and provided motivation.

#### clarity
I find the paper clear in most parts and have not had problems following the main arguments. Related work on learning with similarity measures, indefinite symmetric kernels, and general similarity functions seems to be well covered.

#### quality
I think this paper should really be focusing on learning in reproducing kernel Banach spaces with non-symmetric similarity measures. It is difficult to tell how much novelty it brings compared to relevant related work and how useful it would be in practice. If the authors decide to take this direction, then there should be a detailed experiments section where trade-offs between effectiveness and computational complexity are carefully studied.

It is unclear to me why this work would be associated with attention models and transformers. The story just does not hold and it does not explain the scaled dot-product attention. At the moment, it just seems as an unfinished work that is unnecessarily associated with attention and transformers.

---

> ### Author Response · Authors · 2020-11-16
> **Response to reviewer 1**
>
> Thank you for the review.
>
> In their remark on set kernels, the reviewer seems to have noted another similarity between Transformers and prior work that is complementary to our discussion here. The sketch the reviewer presents, that the output of the Transformer value embedding is a weighted sum of instance embeddings of the source elements, seems to parallel an embedding of a bag in the set kernel terminology.
>
> Our paper focuses on a kernel interpretation of the query and key components of the Transformer, rather than the value embedding component (the value embedding being that which, the reviewer notes, parallels a set kernel). In our view, this query and key pair that produces the attention weights (i.e., the importance weights for the value embedding weighting over the source elements) is the key part of the Transformer. The attention/importance weights are exactly how an attention model can take the target context into account and fixate on the relevant source elements for each of the target elements. The original Vaswani paper was surprising in how these attention weights seemed to, e.g., resolve the noun to which a pronoun was referring.
>
> Our motivation in this paper was to give a mathematical explanation for why the attention weights seemed so powerful. The genesis of this inquiry was the question of why "Luong-style" attention of the query-key dot-product type used in Transformers seemed to perform on par with the "Bahdanau-style" attention that computes attention scores via an MLP: shouldn't the MLP perform better than a linear method? The RKBS story in this paper is the answer at which we arrived.
>
> The representer theorem is presented to make firm the connection between the "Luong-style" attention and kernel methods, as well as to establish the existence and uniqueness of an optimal solution to the Transformer attention calculation, which we feel is not obvious. As we note in the paper, it would not be feasible to use a representer-theorem-based attention weight formula in practice since Transformers tend to deal with much larger datasets than a representer-theorem-based kernel method. Theorem 2 also establishes that this is unnecessary, since the Transformer can approximate this optimal solution arbitrarily well.
>
> The reviewer's note on studying the asymmetry in RKBS learning touches on some of our other interest. In the context of Transformers and attention models, to the best of our knowledge there has not been a rigorous study of the asymmetry between the query and the key embedding routes. Whether, e.g., knowledge of how a query relates to a key could be shared if the direction were to be swapped is an important question in generalization.
>
> We hope we were able to communicate how the RKBS analysis in our paper is motivated by understanding Transformers. Please let us know if there are any concerns we did not address or if you have any additional concerns.

---

> > ### Comment · AnonReviewer1 · 2020-11-24
> > **Response to the Authors**
> >
> > Thank you for the detailed response.
> >
> > I have carefully considered the arguments and the fact that the focus is on explaining the capacity behind the exponential family model responsible for importance weights. Still, my opinion has not changed and I stand with my earlier scores.
> >
> > The work just does not explain the attention mechanism via reproducing kernel Banach spaces, there is no sufficiently rigorous link for this. The part about regularized risk minimization problem in RKBS and Transformers is not directly connected. For example, the considered risk minimization problem is not taken into account when optimizing Transformers. Another weakness in the argument is the fact that the kernel changes with every step of stochastic gradient descent. Namely, the parameters W^Q and W^K are updated and that changes the kernel function. As a result, training of Transformers does not operate in a single RKBS.
> >
> > The claimed contribution boils down to the fact that final (trained) Transformer model will have importance weights that can be associated with some RKBS. This alone, without link to regularized risk minimization problem is not sufficient for me to recommend acceptance.

---

> > > ### Author Response · Authors · 2020-11-25
> > > **Second response to reviewer 1**
> > >
> > > Thank you for taking the time to consider our response. We are glad that we were able to resolve some of the confusion regarding emphasis on the Transformer query/key embedding and attention/importance weights, vs. the Transformer value embedding's set kernel-like operation.
> > >
> > > We agree that there is more room to perform more Banach space-rooted analysis on Transformer-type calculations than what we were able to cover in this paper. The extension from shallow kernel learners like SVMs, which have been well characterized over the years, to deep kernel-like operations like our characterization of Transformers, leads to much of the associated theory needing extension as well. For example, as the reviewer notes in their response and as we discuss in detail in Appendix A, a Transformer-type attention calculation is not yet easily characterizable as a regularized risk minimization problem, since, e.g., the explicit characterization of regularization methods commonly used in deep networks (e.g., dropout) in terms of norm penalties on activations or weights is context-dependent and not fully understood (as we note in our references, authors have been working to characterize dropout in the context of a deep network as a norm penalty for some time). We aim to help fill in some of these gaps, both in the present work and in future work.

---

### Official Review · AnonReviewer3 · 2020-10-28
**My review of paper "Transformers are Deep Infinite-Dimensional Non-Mercer Binary Kernel Machines"**

**Rating:** 6
**Confidence:** 4

**Review:**

In this paper, the authors treat a particular Transformer, "dot-product attention", as  an RKBS kernel called "exponentiated query-key kernel". The explicit form of feature maps and Bach space are given. Moreover, authors term a binary kernel learning problems within the framework of regularized empirical risk minimization. The problem and the correponding representer theorem is new due to its extension to Banach space. A new approximation theorem is also proved and  some experiements are done.
Pros:
The idea of understanding how Transformers work with the help of non-mercer binary kernel  is interesting.
As for the theoretical side, authors provide representer theorem to binary kernel learning for Banach space rather than Hilbert space.

Cons:
The experiment is insufficient because only one dataset is studied.
I think the proof is just a generalization of kernel learning problems on RKBS, without too much difficulty.

---

> ### Author Response · Authors · 2020-11-16
> **Response to reviewer 3**
>
> Thank you for the review. We are glad you found our work interesting.
>
> In response to your concern regarding our experiments being on only one dataset, we have added an experiment on the considerably larger WMT14 English-French translation dataset. These results and updated analysis have been added to the new version of the PDF in Section 6. The results on this dataset seem in agreement with the ones on IWSLT14 English-German, with the "infinite-dimensional" kernels performing the best, and the lower-dimensional ones degrading in performance in the same order.
> We are working on adding some experiments on text classification datasets.
>
> As the reviewer notes, our proof of our representer theorem does draw on earlier works in the RKBS literature. The preliminaries on Banach spaces draw from earlier works and the outline of the proof can be traced back to at least the original RKHS representer theorem proof by Scholkopf et al. However, our problem and theorem are new in the binarization of the kernel learning problem, similar to how the works we mention in our Remark immediately after Definition 3 considered a similar extension to multiple RKHS's. However, as we note in our Appendix A, the weaker conditions on Banach spaces vs Hilbert spaces means we cannot reuse earlier techniques in our setting and needed to create new analysis to handle the Banach space pair. This new analysis is concentrated in Lemmas 2 and 3; perhaps the part in B.3 could be considered fairly straightforward having those lemmas and earlier work.

---

### Author Response · Authors · 2020-11-16
**Some brief notes in addition to individual responses**

Thank you to the reviewers for their thoughtful reviews. We want to make a couple of notes here in addition to the individual responses below.

Most reviewers requested additional experiments. We have added an experiment on the significantly larger WMT14 English-French dataset. After running it, we noticed a mistake we made in configuring our learning rate warmup hyperparameter that may have given the intersection kernel an unfair disadvantage. We are rerunning this now, and as we mention in the individual reviews we are working on adding experiments on some text classification datasets.

We have also added a new remark to the paper (Remark 6 in the new version) that mentions a prior context in which an exponentiated dot product kernel arises.

---

> ### Author Response · Authors · 2020-11-18
> **Results table for WMT14 updated**
>
> The results table in section 6 has been updated reflecting us having corrected our mistake in running the exponentiated intersection kernel trial. The mistake turned out to not have materially affected the results.

---

> > ### Author Response · Authors · 2020-11-25
> > **New version has additional experimental results**
> >
> > Per our prior comments and in response to requests from reviewers for evaluations on more datasets, we have run more experiments with varying the Transformer kernel on text classification, namely sentence sentiment analysis. These results and their discussion have been added to section 6.

---

### Decision · Program_Chairs · 2021-01-07
**Final Decision**

**Decision:**

Reject

**Comment:**

Reviewers have different views on the paper and after going through the reviews, replies and the papers, we believe that
there is room for improvement here.

While the part related to  indefinite symmetric kernels, and general similarity functions seems to be well covered, as
well as the part on Transformers, the relation with learning in RKBS and Transformer is far from being clear and Reviewer 4 makes a strong point on this. For instance,

* what is the goal of the section 5 and Definition 1 . Indeed it is not clear here if the point of the authors is to learn the kernel parameters in equation 9 or to learn to predict the output of a transformer. If it is the latter, the connection with the first part is unclear.

* In Equation 11, I can understand that x and y are the sequences t and s but what is z_ij and how it is obtained? So again, the learning problem drops in without justification and it is not explained how it can be solved. The theoretical results involving the representer theorem is nice though.

* The experiment does not seem very related to the learning problem in Equation 11 introduced by the authors.it seems to me that they are just trying different kernels on top of the dot product.